# An Update on SARS-CoV-2 Clinical Trial Results—What We Can Learn for the Next Pandemic

**DOI:** 10.3390/ijms25010354

**Published:** 2023-12-26

**Authors:** Benediktus Yohan Arman, Juliane Brun, Michelle L. Hill, Nicole Zitzmann, Annette von Delft

**Affiliations:** 1Antiviral Drug Discovery Unit, Oxford Glycobiology Institute, Department of Biochemistry, University of Oxford, Oxford OX1 3QU, UK; juliane.brun@bioch.ox.ac.uk (J.B.); nicole.zitzmann@bioch.ox.ac.uk (N.Z.); 2Kavli Institute for Nanoscience Discovery, University of Oxford, Oxford OX1 3QU, UK; 3Sir William Dunn School of Pathology, University of Oxford, Oxford OX1 3RE, UK; michelle.hill@path.ox.ac.uk; 4Centre for Medicine Discovery, Nuffield Department of Medicine, University of Oxford, Oxford OX3 7BN, UK

**Keywords:** COVID-19, SARS-CoV-2, coronavirus, therapeutics, clinical trial, drug discovery

## Abstract

The coronavirus disease 2019 (COVID-19) pandemic has claimed over 7 million lives worldwide, providing a stark reminder of the importance of pandemic preparedness. Due to the lack of approved antiviral drugs effective against coronaviruses at the start of the pandemic, the world largely relied on repurposed efforts. Here, we summarise results from randomised controlled trials to date, as well as selected in vitro data of directly acting antivirals, host-targeting antivirals, and immunomodulatory drugs. Overall, repurposing efforts evaluating directly acting antivirals targeting other viral families were largely unsuccessful, whereas several immunomodulatory drugs led to clinical improvement in hospitalised patients with severe disease. In addition, accelerated drug discovery efforts during the pandemic progressed to multiple novel directly acting antivirals with clinical efficacy, including small molecule inhibitors and monoclonal antibodies. We argue that large-scale investment is required to prepare for future pandemics; both to develop an arsenal of broad-spectrum antivirals beyond coronaviruses and build worldwide clinical trial networks that can be rapidly utilised.

## 1. Introduction

The coronavirus disease 2019 (COVID-19) pandemic highlighted the world’s under-preparedness when faced with a newly emerging viral pathogen, for which no specific antiviral therapy was available [1]. Coronaviruses that cause human infections include alphacoronaviruses human coronavirus (HCoV) 229E and HCoV NL63, and betacoronaviruses HCoV OC43, HCoV HKU1, severe acute respiratory syndrome coronavirus 1 (SARS-CoV-1), Middle East respiratory syndrome coronavirus (MERS-CoV), and SARS-CoV-2 (Table 1) [2,3,4]. However, over the past two decades, the world has witnessed the emergence of several highly pathogenic betacoronaviruses, with mortality rates ranging from 10% for SARS-CoV-1 in 2002, 34% for MERS-CoV in 2012, and 2% for SARS-CoV-2 in 2019, respectively [5,6]. While the spread of SARS-CoV-1 and MERS-CoV was contained through public health measures, SARS-CoV-2 caused a global pandemic with more than 771.68 million cases recorded and 6.98 million confirmed deaths (Our World in Data, https://ourworldindata.org/grapher/cumulative-deaths-and-cases-covid-19, as of 2 November 2023). SARS-CoV-2 is highly transmissible and can cause asymptomatic or mild symptoms, which makes preventing its spread more difficult [6,7,8]. New variants of SARS-CoV-2 keep emerging, with Omicron being up to 70% more transmissible than previously circulating virus variants [9,10].

At the start of the 2019 pandemic, no therapeutics against SARS-CoV-2 were available. The SARS-CoV-2 life cycle provides insight into antiviral drug targets (Figure 1), with key targets that were exploited for the development of directly acting antivirals (DAA) and host-targeting antivirals (HTA) marked. DAAs target essential viral proteins, such as the main protease (M^Pro^) and RNA-dependent RNA polymerase (RdRp), while host-targeting antivirals inhibit human proteins that the virus utilises for its entry, replication, or assembly.

The choice of drug Is closely linked to the course of natural infection (Figure 2) [11]. DAA and HTA are most suited for use within the first week of infection as they target viral replication. This leads to a short window of opportunity for treatment, as patients often only seek medical attention a few days after symptoms develop [12,13]. In contrast, during later stages of disease, the immune system rather than viral replication as such is the main driver of disease progression, and here, immunomodulatory drugs were shown to be effective [14].

Here, we summarise recent results from randomised controlled trials (RCTs) with linked in vitro results that were performed during the COVID-19 pandemic. We present data on key candidate compounds with a direct impact on virally encoded proteins (DAA), indirect mechanisms of action depriving the virus of essential host factors needed for replication and spread (HTA), and immunomodulatory compounds (Figure 3). An overview of compounds that are now fully approved or under emergency use authorisation (EUA) for COVID-19 by the Food and Drug Administration (FDA) is shown in Table 2. Further, the World Health Organization (WHO) is releasing a living update on COVID-19 therapeutics [15]. In this review, we will summarise clinical trial results of now authorised compounds and highlight other drugs that were shown to be ineffective.

## 2. Directly Acting Antivirals (DAA)

Directly acting antivirals (DAA) that include small molecules and some antibodies, target virally encoded proteins directly and efficiently suppress viral replication in vivo [36] (reviewed in [37,38]). DAA commonly targets viral entry and fusion, or key enzymes like viral polymerases and proteases such as the SARS-CoV2 3-chymotrypsin-like protease (3CL^pro^), also known as main protease (M^pro^), and papain-like protease (PL^pro^) [36,38]. By targeting viral replication directly, DAA are generally thought to have fewer side effects than HTA [39]. However, by targeting the virus directly, DAA are known to cause mutations that can lead to reduced drug efficacy and drug resistance [40,41].

Rapid SARS-CoV-2 small molecule drug discovery efforts were carried out during the 2019 SARS-CoV-2 pandemic, targeting the viral polymerase (remdesivir, Gilead; molnupiravir, Merck, and others) and main protease (nirmatrelvir, Pfizer; ensitrelvir, Shionogi; and others). The pandemic also gave rise to innovative and collaborative approaches such as collaborative alliances across biopharmaceutical companies creating public–private partnerships such as the Innovative Medicines Initiative Corona Accelerated R&D in Europe (https://www.imi-care.eu/, accessed on 8 November 2023), close-knit consortia of companies, and even completely open science consortia with company participation such as the COVID Moonshot [42,43].

### 2.1. Protease Inhibitors

SARS-CoV-2 M^pro^ has been a drug development target from early in the pandemic, with numerous novel SARS-CoV-2 inhibitors in the clinical and preclinical pipeline (Table 3). Two SARS-CoV-2 M^Pro^ inhibitors are now under EUA, including nirmatrelvir/ritonavir (brand name Paxlovid) and ensitrelvir (brand name Xocova).

The covalent SARS-CoV-2 M^pro^ inhibitor nirmatrelvir (PF-07321332) developed by Pfizer is available in combination with ritonavir (available as Paxlovid) and has been evaluated in multiple RCTs to date (Table 3). Nirmatrelvir showed potent antiviral activity in in vitro assays, with a half-maximum effective concentration (EC_50_) of 231 nM in Vero E6 cells [25]. In RCTs, when nirmatrelvir/ritonavir was given within three days of symptoms onset, nirmatrelvir-treated individuals had a significantly reduced risk of severe COVID-19 in comparison to placebo-treated controls, alongside reduced viral loads at day five of treatment [24]. A recent phase 2 RCT study result revealed that nirmatrelvir/ritonavir was more effective than the other leading oral antiviral drug for patients with COVID-19 [44]. A drawback of the combination with the pharmacokinetic (PK) enhancer ritonavir are significant drug-drug interactions, preventing the use in patients with severe renal or hepatic impairment [45,46,47]. A second-generation broad-spectrum protease inhibitor, PF-0781883 has been recently disclosed by Pfizer and is currently in Phase 2 clinical trials, with an initial human PK profile suggesting no further need to co-administer with ritonavir [48].

Ensitrelvir (S-217622) was developed by Shionogi as an oral noncovalent nonpeptidic M^pro^ inhibitor of SARS-CoV-2 using virtual and biological screening of compound libraries and a structural-based drug design strategy of the hit compounds [49,50]. In comparison to nirmatrelvir/ritonavir, ensitrelvir shows an optimized PK profile [50] and displays potent in vitro antiviral activity against SARS-CoV-2 variant of concern (VOC) including Delta (EC_50_ = 34.8 nM) and Omicron BA.1 (EC_50_ = 23.9 nM) variants, as well as other coronaviruses: SARS-CoV-1 (EC_50_ = 0.21 μM), MERS-CoV (EC_50_ = 1.4 μM), and HCoV OC43 (EC_90_ = 0.074 μM) [50,51]. Ensitrelvir is progressing to Phase 2/3 clinical trials. Results disclosed from the Phase 2a trial showed a significant reduction in viral titer and RNA on day four, a median time of negative RT-PCR conversion of two days, and acceptable adverse events [52]. This drug recently received emergency regulatory approval from the Ministry of Health, Labour and Welfare of Japan, with ongoing Phase 3 clinical trials and plans to extend the approval for worldwide use.

Other oral main protease inhibitor candidates in clinical trials include PBI-0451 from Pardes Biosciences and EDP-235 from Enanta Pharmaceuticals [53]. In addition, novel initiatives like the crowdsourced open-science effort COVID Moonshot brought together academic and industrial partners from across the world to develop SARS-CoV-2 M^pro^ inhibitors [43,54]. More than 2400 molecules were synthesised and rapidly shared to create a rich intellectual property-free dataset. The search for a potent covalent COVID-19 M^pro^ inhibitor has also been aided by a computational pipeline to efficiently identify irreversible inhibitors [54]. Due to its open access and free data-sharing nature, the Moonshot fragment dataset may have aided in the early development of the Shionogi M^pro^ compound.

Due to its saliency as a target, repurposing efforts early in the SARS-CoV-2 pandemic focused on the M^pro^. Repurposed protease inhibitors evaluated in RCTs include lopinavir (with and without ritonavir), darunavir, and danoprevir; however, none of the tested inhibitors showed efficacy against SARS-CoV-2 (Table 3).

Lopinavir is a protease inhibitor that has been developed for the treatment of human immunodeficiency virus (HIV) and is used in combination with the PK enhancer ritonavir that blocks the enzyme cytochrome P450 3A [55]. Lopinavir showed some in vitro activity against both SARS-CoV-1 and SARS-CoV-2 with half maximum inhibitory concentrations (IC_50_) of 50 and 26 μM, respectively [55]. Lopinavir/ritonavir was assessed in multiple COVID-19 RCTs, however, it failed to show any clinical benefit (Table 3) and is therefore not recommended as COVID-19 standard care for hospitalised patients [56,57].

Darunavir is another inhibitor of the HIV protease that is in clinical use in combination with the PK enhancer cobicistat [58]. Darunavir was not active against SARS-CoV-2 in vitro at clinically relevant concentrations [59] and in line with these results, did not show clinical efficacy in COVID-19 RCTs (Table 3).

Danoprevir, a hepatitis C virus (HCV) protease (NS3/4A) inhibitor that is used in combination with ritonavir [60] showed cellular activity against SARS-CoV-2 with an EC_50_ of 87 µM in Vero E6 cells [61]. In hospitalised patients infected with SARS-CoV-2, danoprevir with ritonavir shows a good safety profile [60] with shorter times to PCR negativity and shorter hospital stays compared to the lopinavir/ritonavir group [62].

**Table 3 ijms-25-00354-t003:** Small molecule directly acting antivirals (DAA) tested in clinical trials for COVID-19.

Drug Name	Initial Development Target	Mechanism	Trial Attributes	Clinical Trial Settings	Readout	Reference
Lopinavir/ritonavir	HIV	Protease inhibitor	Prospective	Exploratory randomised controlled trial in patients with mild/moderate COVID-19 (n = 86), in Guangzhou in 2020.	Little benefit for improving the clinical outcome of patients compared to no antiviral control.	[63]
Lopinavir/ritonavir	HIV	Protease inhibitor	Prospective	A randomised, controlled, open-label, platform trial in patients with COVID-19, admitted to 176 hospitals (n = 1616) in the United Kingdom in 2020 (RECOVERY trial).	Treatment was not associated with reductions in 28-day mortality, the duration of hospital stays, or the risk of progressing to invasive mechanical ventilation or death.	[56]
Lopinavir/ritonavir	HIV	Protease inhibitor	Prospective	A randomised, controlled, open-label trial in hospitalised adult patients (n = 199) with confirmed SARS-CoV-2 infection in Hubei, China in 2020 (Lopinavir Trial for Suppression of SARS-CoV-2 in China/LOTUS China).	No benefit was observed with treatment compared to the standard care.	[64]
Lopinavir, ritonavir, ribavirin, and interferon beta-1b combination	HIV	Protease inhibitor	Prospective	A multi-centre, prospective, open-label, randomised, phase 2 trial in adults (n = 127) with COVID-19 at 6 hospitals in Hong Kong in 2020, comparing lopinavir, ritonavir, ribavirin, and interferon beta-1b combination group and lopinavir and ritonavir control group.	The combination group was safe and superior to lopinavir-ritonavir alone in alleviating symptoms and shortening the duration of viral shedding and hospital stay in patients with mild to moderate COVID-19.	[65]
Lopinavir (without interferon)	HIV	Protease inhibitor	Prospective	A randomised trial with the intention to treat primary analysis in hospitalised patients with COVID-19. The study was conducted at 405 hospitals in 30 countries (n = 11,330) in 2020 (WHO SOLIDARITY trial).	Little or no effect is marked by no reduction in mortality, initiation of ventilation, or hospitalization duration.	[66]
Lopinavir/ritonavir	HIV	Protease inhibitor	Retrospective observational study	A multi-centre study to test the association of risk factors and therapies with in-hospital COVID-19 mortality (n = 3451) in Italy in 2020 (CORIST study).	No change in the death rate after treatment with lopinavir-ritonavir.	[67]
Lopinavir/ritonavir	HIV	Protease inhibitor	Prospective	A randomised trial in critically ill patients from different countries (n = 694) in 2020 (REMAP-CAP study).	Worsened outcomes compared to no antiviral therapy.	[68]
Lopinavir/ritonavir, lopinavir/ritonavir-interferon (IFN)-β-1a	HIV	Protease inhibitor	Prospective	Multi-centre, open-label, randomised, adaptive, controlled trial in COVID-19 inpatients (n = 603) requiring oxygen and/or ventilatory support, conducted at 30 sites in France and Luxembourg in 2020 (DisCoVeRy trial).	No improvement in clinical status at day 15 or SARS-CoV-2 clearance in respiratory tract specimens compared to the standard care.	[69]
Danoprevir with ritonavir booster	HCV	Protease inhibitor	Prospective	An open-label, single-arm study on treatment of naïve and experienced COVID-19 patients (n = 11, China, 2020) for the first time with the rate of composite adverse outcomes as the primary endpoint.	No composite adverse outcomes.	[60]
Danoprevir	HCV	Protease inhibitor	Prospective	A comparative study between two treatment groups, danoprevir and lopinavir/ritonavir treatment (n = 33) in Nanchang, China in 2020.	Shorter duration of time to achieve negative nucleic acid testing and hospital stay compared to the lopinavir/ritonavir group.	[62]
Darunavir/cobicistat	HIV	Protease inhibitor	Prospective	A single-centre, randomised, and open-label trial in mild patients with PCR-confirmed COVID-19 (n = 30) in Shanghai, China in 2020.	No increase in the proportion of conversion compared to the interferon alpha 2b and standard care control group.	[70]
Darunavir/cobicistat	HIV	Protease inhibitor	Retrospective observational study	A multi-centre study aimed at testing the association of risk factors and therapies with in-hospital COVID-19 mortality (n = 3451) in Italy in 2020 (CORIST study).	No benefit was observed with treatment compared to standard care.	[67]
Darunavir/ritonavir in combination with hydroxychloroquine	HIV	Protease inhibitor	Prospective	Open-labelled, randomised controlled trial with an intention to treat protocol (n = 113) in Thailand from December 2020 to April 2021.	No virological or clinical benefit was observed.	[71]
Darunavir/cobicistat	HIV	Protease inhibitor	Retrospective	Multi-centre observational study in Qatar in 2020 on adult patients (n = 400) hospitalised due to COVID-19.	Less time for clinical improvement compared to treatment with lopinavir-ritonavir.	[72]
Nirmatrelvir/ritonavir	SARS-CoV-2/COVID-19	Protease inhibitor	Prospective	A phase 2–3 double-blind, randomised, controlled multi-centre trial in 21 countries, in symptomatic, unvaccinated, non-hospitalised adults (n = 2246) at high risk for progression to severe COVID-19, conducted in 2021 in 21 countries.	Lower risk of progression to severe COVID-19 compared to the placebo control.	[24]
Nirmatrelvir/ritonavir	SARS-CoV-2/COVID-19	Protease inhibitor	Retrospective	A multi-centre (11 hospitals and 55 clinics in Minnesota, USA), retrospective review of patients evaluated for a diagnosis of COVID-19 (n = 66,007) in 2020–2022.	Prevalent medical contraindications to nirmatrelvir/ritonavir.	[47]
Nirmatrelvir/ritonavir	SARS-CoV-2/COVID-19	Protease inhibitor	Retrospective	An accelerated, randomised, double-blind, placebo-controlled, phase I study, conducted in Japanese cohort in 2021.	Significant decrease in SARS-CoV-2 viral load in patients and prevention of severe disease, hospitalisation, and death.	[73]
Nirmatrelvir/ritonavir	SARS-CoV-2/COVID-19	Protease inhibitor	Prospective	An open-label, multi-centre, randomised controlled trial in hospitalised adult patients (n = 264) with severe COVID-19 comorbidities at 5 hospitals in Shanghai, China in 2022.	No significant reduction in the risk of all-cause mortality on day 28 and the duration of SARS-CoV-2 RNA clearance.	[74]
Nirmatrelvir/ritonavir	SARS-CoV-2/COVID-19	Protease inhibitor	Prospective	An open-label, multi-centre, randomised, controlled, phase 2 adaptive pharmacometrics platform trial in adult patients (n = 383) with early symptomatic COVID-19 in Thailand, Brazil, Pakistan, and Laos in 2022 (PLATCOV study).	Ritonavir-boosted nirmatrelvir accelerated oropharyngeal SARS-CoV-2 viral clearance with a significantly greater effect than molnupiravir.	[44]
Ensitrelvir	SARS-CoV-2/COVID-19	Protease inhibitor	Prospective	A multi-centre, double-blind, phase 2a part of a phase 2/3 study in mild to moderate patients with COVID-19 (n = 69), conducted in Japan from September 2021 to January 2022.	Reduction of SARS-CoV-2 RNA on day 4 and decrease in the median time to the infectious viral clearance compared to placebo.	[52]
Ensitrelvir	SARS-CoV-2/COVID-19	Protease inhibitor	Prospective	A multi-centre, double-blind, phase 2b part of a phase 2/3 study in mild to moderate patients with COVID-19 (n = 341) in Japan and South Korea in 2022.	Favorable antiviral efficacy (as a change from baseline in SARS-CoV-2 titer on day 4) and potential clinical benefit with an acceptable safety profile compared to placebo.	[75]
Ensitrelvir	SARS-CoV-2/COVID-19	Protease inhibitor	Prospective	A multi-centre, randomised, double-blind, placebo-controlled, phase 3 study in mild to moderate patients with COVID-19 (Phase 3 part) in Japan, Korea, Singapore, and Vietnam, started in 2022.	Reduced time to the resolution of 5 symptoms in patients with mild-to-moderate COVID-19 compared to placebo.	[76]
Ensitrelvir	SARS-CoV-2/COVID-19	Protease inhibitor	Prospective	A phase 1, multi-centre, single-arm, open-label study in healthy Japanese adult participants (n = 42) in 2022 to investigate the effect of ensitrelvir on the pharmacokinetics of CYP3A substrates and assess the pharmacokinetics, safety, and tolerability following multiple-dose administration.	Ensitrelvir at the clinical dose was well tolerated with no additional safety signal and can be co-administered with several CYP3A substrates likely to be used in COVID-19 patients.	[77]
Favipiravir	Influenza virus	RdRp inhibitor	Prospective	A randomised, open-label trial of early versus late therapy in adolescent and adult patients with COVID-19 (n = 89) at 25 hospitals in Japan in 2020.	No improvement in viral clearance by day 6, although there was a numerical reduction in time to defervescence.	[78]
Favipiravir	Influenza virus	RdRp inhibitor	Prospective	An adaptive, multi-centre, open-label, randomised, Phase 2/3 clinical trial compared to the standard of care in hospitalised patients with moderate COVID-19 pneumonia (n = 60) at 6 sites in Russia in 2020.	Rapid antiviral response against SARS-CoV-2 shown by the higher proportion of patients who achieved negative PCR on Day 5 when treated using favipiravir compared to the control group.	[79]
Favipiravir	Influenza virus	RdRp inhibitor	Prospective	A randomised, open-label, parallel-arm, multi-centre, phase 3 trial in adults (18–75 years) patients with RT-PCR confirmed COVID-19 and mild-to-moderate symptoms (including asymptomatic) (n = 150) in India in 2020.	No statistically significant result on the primary endpoint of time to RT-PCR negativity.	[80]
Favipiravir combined with inhaled interferon beta-1β	Influenza virus	RdRp inhibitor	Prospective	A randomised, open-label controlled trial in adults hospitalised with moderate to severe COVID-19 pneumonia (n = 89) in Oman in 2020.	No difference in clinical outcome compared to the standard arm hydroxychloroquine.	[81]
Favipiravir	Influenza virus	RdRp inhibitor	Prospective	A randomised exploratory trial in hospitalised adult patients with COVID-19 (n = 30) in Zhejiang, China in 2020.	No clinical improvement compared to standard care.	[82]
Favipiravir	Influenza virus	RdRp inhibitor	Prospective	A phase 2, double-blind, randomised, controlled outpatient trial in asymptomatic or mildly symptomatic adults with a positive SARS-CoV-2 RT-PCR within 72 h of enrolment (n = 149) in California, USA from July 2020 to March 2021.	No clinical benefit shown as no difference in shedding cessation time, and time to symptom resolution, compared to placebo.	[83]
Favipiravir	Influenza virus	RdRp inhibitor	Prospective	A phase 2, randomised placebo-controlled phase 2 trial of favipiravir versus matched placebo in individuals infected with COVID-19 and 5 days or less of symptoms (n = 199) in Australia from July 2020 to September 2021.	No improvement in the time to virological cure or clinical outcomes and no evidence of an antiviral effect in early symptomatic COVID-19 infection.	[84]
Favipiravir	Influenza virus	RdRp inhibitor	Prospective	A phase 3, multi-centre, open-label, randomised controlled trial of oral favipiravir in adult patients (n = 499) who were newly admitted to hospitals with proven or suspected COVID-19 across 5 sites in the UK, Brazil, and Mexico in 2020–2021 (PIONEER trial).	No significant difference in time to recovery and serious adverse events.	[85]
Favipiravir	Influenza virus	RdRp inhibitor	Prospective	A triple-blind, randomised, placebo-controlled trial in mild to moderate adult patients in an outpatient setting (n = 77) in Iran from December 2020 to March 2021.	No reduction in hospitalisation rate.	[86]
Remdesivir	EBOV	RdRp inhibitor	Prospective	A double-blind, randomised, placebo-controlled trial in adults who were hospitalised with COVID-19 and had evidence of lower respiratory tract infection (n = 1062) in the United States, Denmark, UK, Greece, Germany, Korea, Mexico, Spain, Japan, and Singapore in 2020 (ACTT-1 study).	A shorter time to recovery was observed in patients treated with remdesivir compared to placebo.	[16]
Remdesivir	EBOV	RdRp inhibitor	Prospective	A randomised, open-label, phase 3 trial involving hospitalised patients with severe COVID-19 (n = 397) at 55 hospitals in the United States, Italy, Spain, Germany, Hong Kong, Singapore, South Korea, and Taiwan in 2020.	No significant difference between 5-day and 10-day courses, and no clinical benefit compared to placebo control.	[87]
Remdesivir	EBOV	RdRp inhibitor	Prospective	A randomised, open-label trial in hospitalised patients (n = 596) with moderate COVID-19 in the United States, Europe, and Asia in 2020.	No significant clinical status difference at 11 days compared to the standard care.	[88]
Remdesivir	EBOV	RdRp inhibitor	Prospective	A randomised, double-blind, placebo-controlled, multi-centre trial in hospitalised adults with severe COVID-19 (n = 237) in Wuhan, Hubei, China in 2020.	No significant difference in time to clinical improvement compared to placebo.	[89]
Remdesivir	EBOV	RdRp inhibitor	Prospective	A randomised, multi-centre, open control, in hospitalised adult patients with a diagnosis of COVID-19 (n = 11,330) at 405 hospitals in 30 countries in 2020 (WHO Solidarity trial).	Little or no effect marked by no reduction in mortality before or after day 28, initiation of ventilation, or hospitalization duration.	[66]
Remdesivir	EBOV	RdRp inhibitor	Prospective	A randomised, double-blind, placebo-controlled trial involving non-hospitalised patients with COVID-19 who had symptom onset within the previous 7 days and who had at least one risk factor for disease progression (age ≥ 60 years, obesity, or certain coexisting medical conditions) (n = 562), at 64 sites in the United States, Spain, Denmark, and the United Kingdom, September 2020 to April 2021 (PINETREE study).	Remdesivir, given in a 3-day course, showed an acceptable safety profile and resulted in an 87% lower risk of hospitalization or death compared to placebo.	[17]
Remdesivir-dexamethasone	EBOV	RdRp inhibitor	Prospective	A controlled non-randomised study in COVID-19 patients requiring supplemental oxygen therapy (n = 151) in Italy in 2021.	Significant reduction in mortality, length of hospitalization, and faster SARS-CoV-2 clearance, compared to dexamethasone alone.	[90]
Molnupiravir	VEEV	RdRp inhibitor	Prospective	A phase Ib/IIa, dose-escalating, open-label, randomised, controlled trial in adult outpatients with PCR-confirmed SARS-CoV-2 infection within 5 days of symptom onset (n = 18) in the United Kingdom in 2020.	Molnupiravir is safe and well-tolerated.	[91]
Molnupiravir	VEEV	RdRp inhibitor	Prospective	A randomised, controlled trial in patients with mild or moderate COVID-19 (n = 116) in Shenzen, China in March 2022.	Acceleration of SARS-CoV-2 Omicron variant RNA clearance in patients, compared to the standard of care.	[92]
Molnupiravir	VEEV	RdRp inhibitor	Prospective	A phase 2a double-blind, placebo-controlled, randomised, multi-centre clinical trial in unvaccinated participants with confirmed SARS-CoV-2 infection and symptom duration of less than 7 days (n = 204) in the United States from June 2020 to January 2021.	Acceleration of viral RNA and infectious virus clearance, compared to placebo control.	[93]
Molnupiravir	VEEV	RdRp inhibitor	Prospective	A randomised, double-blind, placebo-controlled, multi-centre phase 3 trial in non-hospitalised adults with mild to moderate COVID-19 (n = 1433) at 107 sites globally in 2020—2022 (MOVe-OUT study).	Reduction of hospitalisation duration and death compared to the placebo control group.	[94]
Molnupiravir	VEEV	RdRp inhibitor	Prospective	A phase 3, double-blind, randomised, placebo-controlled trial in non-hospitalised, unvaccinated adults with mild-to-moderate, laboratory-confirmed COVID-19 and at least one risk factor for severe COVID-19 (n = 1433) in 2021, globally (MOVe-OUT study).	Reduction of the risk of hospitalization or death compared to the placebo group.	[22]
Molnupiravir	VEEV	RdRp inhibitor	Prospective	A randomised, placebo-controlled, double-blind, phase 2 trial in adult (aged ≥ 18 years) outpatients with PCR-confirmed, mild-to-moderate SARS-CoV-2 infection (n = 180) in the United Kingdom in 2020 (AGILE CST-2 study).	Inconclusive evidence of antiviral activity in vaccinated and unvaccinated patients.	[95]
Molnupiravir	VEEV	RdRp inhibitor	Prospective	A multi-centre, open label, multigroup, platform adaptive randomised controlled trial involving 26,411 participants in the community in the United Kingdom from December 2021 to April 2022 (PANORAMIC trial study).	No reduction in the frequency of COVID-19-associated hospitalizations or deaths among high-risk vaccinated adults.	[96]
Molnupiravir	VEEV	RdRp inhibitor	Prospective	A phase 3, randomised, placebo-controlled trial in non-hospitalised at-risk adults with mild-to-moderate COVID-19 and immunocompromised status (n = 55) in 2021, globally (MOVe-OUT study).	Increased clearance of infectious virus compared to the placebo control, efficacious in immunocompromised patients.	[97]
Azvudine/FNC	HIV	RdRp inhibitor	Prospective	A randomised, open-label, controlled trial comparing azvudine and symptomatic treatment (FNC group, n = 10) to standard antiviral and symptomatic treatment (control group, n = 10) in mild COVID-19 patients in Guangshan, China in 2020.	Shorter duration of the mean times of the first nucleic acid negative conversion.	[98]
Azvudine/FNC	HIV	RdRp inhibitor	Retrospective	A single-centre, retrospective cohort study in hospitalised patients (n = 245 for each Azvudine and matched control groups) from December 2022 to January 2023 in Hunan, China.	Substantial clinical benefits in composite disease progression outcome of hospitalised patients with COVID-19 and pre-existing conditions.	[99]
Azvudine/FNC	HIV	RdRp inhibitor	Retrospective	A retrospective, real-world clinical study in hospitalised COVID-19 patients comparing azvudine (n = 281) with nirmatrelvir-ritonavir (n = 281) from December 2022 to January 2023 in Hunan, China.	A lower crude incidence rate of composite disease progression outcome compared to the nirmatrelvir-ritonavir group.	[100]
Sofosbufir/daclatasvir and ribavirin	HCV	RdRp inhibitor	Prospective	A single-centre randomised controlled trial in adults with moderate COVID-19 (n = 48) in Iran in 2020.	No difference between the experimental group compared to the standard care control.	[101]
Sofosbuvir/daclatasvir	HCV	RdRp inhibitor	Prospective	A single-centre, open-label, parallel trial in hospitalised patients (n = 62) with severe COVID-19 in Iran in 2020.	No decrease in mortality compared to the ribavirin control group.	[102]
Sofosbuvir/daclatasvir	HCV	RdRp inhibitor	Prospective	An open-label, multi-centre, randomised, controlled clinical trial in adults with moderate or severe COVID-19 (n = 66) in Iran in 2020.	A reduced duration of hospital stay compared to standard care.	[103]
Sofosbuvir/ledipasvir	HCV	RdRp inhibitor	Prospective	A single-centre, open-label, randomised clinical trial in patients with mild to moderate COVID-19 (n = 82) in Iran in 2020.	No difference in clinical response, duration of hospital and ICU stay, or 14-day mortality compared to the standard care control.	[104]
Sofosbuvir/daclatasvir with hydroxychloroquine	HCV	RdRp inhibitor	Prospective	A randomised, controlled, single-centre clinical trial in outpatients with mild COVID-19 (n = 55) in Iran in 2020.	No clinical symptom alleviation compared to the hydroxychloroquine-only control group.	[105]
Sofosbuvir/daclatasvir	HCV	RdRp inhibitor	Prospective	A placebo-controlled, double-blind, randomised clinical trial in adults hospitalised with COVID-19 (n = 1083) at 19 hospitals in Iran in 2020.	No effect on patients’ hospital discharge or survival compared to the placebo control.	[106]
Sofosbuvir/ledipasvir	HCV	RdRp inhibitor	Prospective	A single-blinded parallel-randomised controlled trial of patients (n = 250) treated with sofosbuvir/ledipasvir in the intervention group and oseltamivir, hydroxychloroquine, and azithromycin (OCH group) in the control group, conducted in Egypt in 2020.	Reduced time to patients’ recovery and time of hospital stay compared to the control group.	[107]

FNC, 2-deoxy-2-β-fluoro-4-azidocytidine; EBOV, Ebola virus; HIV, human immunodeficiency virus; HCV, hepatitis C virus; RdRp, RNA-dependent RNA polymerase; VEEV, Venezuelan equine encephalitis virus.

### 2.2. RNA-Dependent RNA Polymerase (RdRp) Inhibitor

The RNA-dependent RNA polymerase (RdRp) is essential for viral replication and is highly conserved among coronaviruses and positive-strand RNA viruses, and a clinically validated target across many viruses [108,109]. Several existing RdRp inhibitors developed against other RNA viruses, including remdesivir [110,111], molnupiravir [112], and favipiravir [113], were in late-stage development or clinical use at the start of the pandemic and showed promising in vitro activity against the SARS-CoV-2. Subsequently, numerous clinical trials were conducted to explore their potential against SARS-CoV-2 infection (Table 3).

Remdesivir is a nucleotide analogue inhibitor of the RdRp that was originally developed for the Ebola virus (EBOV) [114]. It shows potent broad-spectrum antiviral activity against pathogenic animal and human coronaviruses in vitro [110,111,115]. However, remdesivir’s intravenous administration limits its widespread use in the community. Based on multiple clinical trials, remdesivir was the first antiviral drug approved for the treatment of COVID-19 in adults and paediatric patients (≥28 days old and weighing ≥ 3 kg) with a positive SARS-CoV-2 test, for both hospitalised patients, and non-hospitalised individuals with mild-to-moderate COVID-19 that are at high risk for progression to severe COVID-19 [116]. One randomised, double-blind, placebo-controlled clinical trial (ACTT-1 study) showed a shorter median time to recovery for remdesivir-treated patients compared to placebo [16]. Two additional trials sponsored by Gilead Sciences informed the approval. An open-label clinical trial of hospitalised adults with moderate COVID-19 showed an improvement in symptoms in patients receiving a five-day course of remdesivir, however, no improvement was demonstrated in those receiving a 10-day course of remdesivir [88]. A third randomised, open-label trial in hospitalised adults with severe COVID-19 showed no statistically significant differences in recovery or mortality rates [87]. However, it is important to note that with DAA, initiating treatment early whilst viral loads are high is crucial for compound efficacy (Figure 2), and variable treatment windows may explain conflicting clinical trial results [17]. An oral pro-drug of remdesivir, obeldesivir, shows excellent cross-reactivity against multiple coronaviruses including MERS, in vitro [117]. A recent Phase 3 RCT to evaluate the efficacy and safety for the treatment of COVID-19 in non-hospitalised patients with a high risk for disease progression was discontinued due to lower than expected incidence rates and related hospitalisations or all-cause death, whilst a Phase 3 RCT in hospitalised patients is still ongoing [118].

Molnupiravir is an orally available RdRp inhibitor with broad-spectrum antiviral activity [112] and was initially developed as an oral antiviral against Venezuelan equine encephalitis virus (VEEV) [108]. It acts as a nucleoside analogue in the RNA elongation process where it incorporates mutation errors that accumulate into an “error catastrophe” and subsequent virus replication failure [22,108,112]. Molnupiravir is authorised for emergency use for SARS-CoV-2 infection by the FDA (Table 2) in adults with confirmed mild-to-moderate SARS-CoV-2 infection, including those who are at high risk for progression to severe disease and those for whom alternative treatment options are not accessible or clinically appropriate. The FDA authorisation was based on the randomised, double-blind, placebo-controlled MOVe-OUT trial, demonstrating a reduction in hospitalization and mortality in the molnupiravir-treated group compared to the placebo [97]. However, the European Medicines Agency refused market authorisation for molnupiravir [119], arguing that it was not possible to conclude that molnupiravir reduces the risk of hospitalisation or death in adults at risk of severe disease.

Several other RdRp inhibitors that have been developed for other viral infections have been investigated for clinical efficacy in COVID-19 as part of intensive repurposing efforts. Favipiravir is a guanine analogue that selectively inhibits the RdRp and has been developed as a novel antiviral compound against influenza by Toyama Chemical Co. Favipiravir has been evaluated against SARS-CoV-2 infection in multiple clinical trials (Table 3), with no significant effects on hospitalisation and mortality even when administered within 5 days of symptoms developing [84]. Sofosbuvir, a potent inhibitor developed against the RdRp of HCV, has and is in clinical use in combination for the treatment of chronic HCV infection in combination with ribavirin, ledipasvir, and daclatasvir for different genotypes. In multiple RCTs conducted with sofosbuvir in SARS-CoV-2 infection (Table 3), no effect on endpoints was observed, apart from a small open-label study demonstrating an effect on median hospital stay [103]. Azvudine, a nucleoside analogue RdRp inhibitor originally developed against HIV infection [120,121] has been approved for the treatment of COVID-19 by the Chinese regulatory agency in 2022, citing a phase 3 clinical trial showing “improved clinical symptoms”, compared to a placebo [122,123]. However, detailed clinical trial data has not been published to date. Publicly available data is limited, with studies available showing an impact on SARS-CoV-2 viral replication and a shortened time to viral clearance in patients with mild COVID-19, compared to the standard antiviral treatment [98], as well as an impact on disease progression outcome in retrospective studies (Table 3).

### 2.3. Monoclonal Antibodies

Neutralising monoclonal antibodies, mostly targeting the receptor binding domain (RBD) of the SARS-CoV-2 spike protein, has been tested in several clinical trials [38] and are reviewed elsewhere [124] (Table 4). They offer a highly effective therapy against viral infections. However, their use is limited in clinical practice due to (i) the need for intravenous application which limits their use to hospital settings [125,126]; (ii) high costs that restrict access for large parts of the global population [127]; (iii) the reduced efficacy of some antibodies against VoCs [128]. Specifically, some antibodies with good efficacy against earlier SARS-CoV-2 variants were less potent against Omicron and its sub-variants [129]. Antibodies are effective in mild to moderate disease if given early. Further developments in the use and applicability of monoclonal antibodies for preventing COVID-19 and for early therapy are underway, including alternative applications such as nasal sprays [130].

Examples of monoclonal antibodies developed against SARS-CoV-2 include the antibody cocktail bamlanivimab with etesevimab from Eli Lilly and Company [131], Regeneron’s REGN-COV2 casirivimab and imdevimab cocktail [125,132], and the monotherapy sotrovimab by Vir Biotechnology in collaboration with GlaxoSmithKline [133].

The combination of bamlanivimab and etesevimab lowers COVID-19-related hospitalisation and mortality in mild to moderate COVID-19 cases and reduces SARS-CoV-2 viral load [134]. The use of antibody cocktails that target different neutralisation epitopes may be one way to maintain their therapeutic efficacy [135]. The non-competing monoclonal antibodies casirivimab and imdevimab bind to two different sites on the RBD of the SARS-CoV-2 spike protein resulting in the blockage of viral entry into host cells [125]. This antibody combination therapy reduced mortality at 28 days in seronegative patients but not in seropositive patients at baseline [125].

The neutralising antibody sotrovimab, developed by GlaxoSmithKline and Vir Biotechnology neutralises SARS-CoV-2 by targeting a highly conserved epitope in the RBD of the spike protein [126]. Treatment with sotrovimab reduced the risk of COVID-19 disease progression to hospitalisation or death in mild to moderate high-risk patients [126]. Sotrovimab monotherapy retained activity against SARS-CoV-2 VOCs including Omicron [133], however, has been associated with the rapid development of spike gene mutations in vitro [128] and in vivo [133].

Bebtelovimab, developed by Eli Lilly, is a neutralising IgG1 monoclonal antibody that binds to an epitope within the RBD of the SARS-CoV-2 S protein with broad neutralising activity to all SARS-CoV-2 VOCs, including the Omicron variant [136]. Bebtelovimab has been studied in the BLAZE-4 clinical trial and is associated with greater viral clearance. It is effective for the treatment of mild to moderate COVID-19 in adults and children above 12 years old who are at risk of disease progression and hospitalisation [137].

The long-lasting monoclonal antibody combination of tixagevimab and cilgavimab (brand name Evusheld from AstraZeneca) was derived from antibodies isolated from B cells of patients infected with SARS-CoV-2 [138,139].

The FDA removed its EUA for several monoclonal antibodies including bebtelovimab, tixagevimab-cilgavimab, sotrovimab, bamlanivimab-etesevimab, and casirivimab-imdevimab [34], due to their reduced efficacy against Omicron and its subvariants that are now circulating at high frequency [128,129].

**Table 4 ijms-25-00354-t004:** Biologicals directly acting antivirals (DAA) tested in clinical trials for COVID-19.

Drug Name	Initial Development Target	Mechanism	Trial Attributes	Clinical Trial Settings	Readout	Reference(s)
LY-CoV555	SARS-CoV-2	Anti-spike RBD neutralising antibody	Prospective	A randomised, placebo-controlled phase 2 trial involving outpatients with recently diagnosed mild or moderate COVID-19 (n = 452) at 41 centers in the United States in Sep 2020 (Blocking Viral Attachment and Cell Entry with SARS-CoV-2 Neutralising Antibodies, BLAZE-1 trial).	The accelerated natural decline in viral load over time in one of three doses, compared to placebo.	[140]
LY-CoV555 coadministered with remdesivir	SARS-CoV-2	Anti-spike RBD neutralising antibody	Prospective	A randomised, placebo-controlled trial in hospitalised patients who had COVID-19 without end-organ failure (n = 314) at 31 sites in the United States, Denmark, and Singapore from August to October 2020 (ACTIV-3/TICO LY-CoV555 study)	No efficacy compared to the placebo group.	[141]
Bamlanivimab monotherapy	SARS-CoV-2	Anti-spike neutralising antibody	Prospective	A randomised, double-blind, sponsor unblinded, placebo-controlled, single ascending dose in hospitalised patients with COVID-19 (n = 26) at 5 sites in the United States from May to June 2020.	Favorable safety profile was demonstrated.	[142]
Bamlanivimab monotherapy	SARS-CoV-2	Anti-spike neutralising antibody	Prospective	A randomised, placebo-controlled phase 2/3 trial in ambulatory patients who tested positive for SARS-CoV-2 infection and had one or more mild to moderate symptoms (n = 613) at 49 centers in the United States from June to October 2020 (BLAZE-1 trial).	No significant difference in viral load reduction.	[143]
Bamlanivimab and etesevimab	SARS-CoV-2	Anti-spike neutralising antibody	Prospective	A randomised, placebo-controlled phase 2/3 trial in ambulatory patients who tested positive for SARS-CoV-2 infection and had one or more mild to moderate symptoms (n = 613) at 49 centers in the United States from June to October 2020 (BLAZE-1 trial).	Statistically significant reduction in SARS-CoV-2 viral load at day 11 compared to placebo control.	[143]
Bamlanivimab	SARS-CoV-2	Anti-spike neutralising antibody	Prospective	A randomised, double-blind, single-dose, phase 3 trial among residents and staff in skilled nursing and assisted living facilities (n = 1175) from August to November 2020 at 74 facilities in the United States.	The reduced incidence of COVID-19 among residents and staff.	[144]
Bamlanivimab and etesevimab	SARS-CoV-2	Anti-spike neutralising antibody	Prospective	A randomised, placebo-controlled phase 3 trial in a cohort of ambulatory patients with mild or moderate COVID-19 who were at high risk for progression to severe disease (n = 769) in the United States in 2021 (BLAZE-1 trial).	Lower incidence of COVID-19-related hospitalization and death and acceleration of the decline in the SARS-CoV-2 viral load, compared to placebo.	[134,145]
Bamlanivimab	SARS-CoV-2	Anti-spike neutralising antibody	Prospective	A multi-centre, randomised, placebo-controlled trial in hospitalised patients with COVID-19 without end-organ failure (n = 314) in the United States from August to October 2020 (ACTIV-3/TICO Bamlanivimab Study).	No firm conclusion was achieved due to the limited sample size and the efficacy and safety of bamlanivimab may differ depending on the status of endogenous neutralising antibody response.	[146]
Bamlanivimab	SARS-CoV-2	Anti-spike neutralising antibody	Prospective	A randomised, placebo-controlled in non-hospitalised adults with early variants of SARS-CoV-2 using 7000 mg (n = 94) and 700 mg (n = 223) dose cohorts from August to November 2020 in the United States.	Evidence of faster reductions in nasopharyngeal SARS-CoV-2 RNA levels but not shorter symptom durations.	[147]
Casirivimab and imdevimab (REGEN-COV)	SARS-CoV-2	Anti-spike RBD antibody	Prospective	A randomised, placebo-controlled trial in persons at high risk for infection because of household exposure to a person with SARS-CoV-2 infection (n = 2475) at 112 sites in the United States, Romania, and Moldova in 2020.	Subcutaneous injection prevented symptomatic COVID-19 and asymptomatic SARS-CoV-2 infection in previously uninfected household contacts of infected persons. Among the participants who became infected a reduced duration of symptomatic disease and high viral load were recorded.	[148]
Casirivimab and imdevimab (REGEN-COV)	SARS-CoV-2	Anti-spike RBD antibody	Prospective	A randomised, placebo-controlled phase 3 trial in outpatients with COVID-19 and risk factors for severe disease (n = 3088) in the United States from September 2020 to January 2021.	The reduced risk of COVID-19-related hospitalization or death from any cause, resolution of symptoms, and reduced SARS-CoV-2 viral load more rapidly than placebo.	[132]
Casirivimab and imdevimab (REGN-COV2)	SARS-CoV-2	Anti-spike RBD antibody	Prospective	A randomised, placebo-controlled phase 1–3 trial in outpatients (n = 275) in the United States from June to August 2020.	Reduction of viral load compared to placebo control.	[149]
Casirivimab and imdevimab	SARS-CoV-2	Anti-spike RBD antibody	Prospective	A multi-centre, randomised, double-blind, placebo-controlled phase 3 trial in close household contacts of SARS-CoV-2-infected people (n = 314) in the United States, Romania, and Moldova in 2020.	The reduced incidence of symptomatic COVID-19 over 28 days.	[150]
Casirivimab and imdevimab	SARS-CoV-2	Anti-spike RBD antibody	Prospective	A randomised, controlled, open-label platform trial involving 9785 patients admitted to hospital with COVID-19 at 127 hospitals in the United Kingdom from September 2020 to May 2021 (RECOVERY study).	The reduced 28-day mortality in patients who were seronegative at baseline but not in those who were seropositive at baseline.	[125]
Vilobelimab	Avian influenza (H7N9)	Anti C5a monoclonal antibody that inhibits the complement system activation	Prospective	A randomised, double-blind, placebo-controlled, multi-centre phase 3 trial in adult hospitalised patients who were receiving invasive mechanical ventilation and a confirmed SARS-CoV-2 infection (n = 368) at 46 hospitals in the Netherlands, Germany, France, Belgium, Russia, Brazil, Peru, Mexico, and South Africa from October 2020 to October 2021 (PANAMO study).	Improvement in the survival of invasive mechanically ventilated patients with COVID-19 and a significant decrease in mortality.	[19]
Vilobelimab	Avian influenza (H7N9)	Anti C5a monoclonal antibody that inhibits the complement system activation	Prospective	An exploratory, open-label, randomised phase 2 trial in patients with severe COVID-19 pneumonia (n = 17) at 46 hospitals in the Netherlands, Germany, France, Belgium, Russia, Brazil, Peru, Mexico, and South Africa from March to April 2020.	Beneficial effect in COVID-19 patients, shown by the decrease in inflammatory response and hypercoagulability.	[151]

## 3. Host-Targeting Antivirals (HTA)

Host-targeting antivirals (HTA) are drugs that modify host cell pathways required for viral replication [152]. They often modulate virus-host interactions by targeting human proteins used by viruses. As they may target host cell pathways used by several viral families, HTAs may be suitable as a first-line antiviral drug when a novel virus emerges, provided the novel virus relies on the same host pathway. Conceptually, HTAs could be given early in a pandemic, prior to emerging viruses being characterised at the molecular level, and may carry a lower risk of drug resistance [40,41]. Even though no HTA are approved for COVID-19 therapy due to the lack of efficacy in RCTs performed to date, we discuss results on a range of clinically tested compounds.

### 3.1. Inhibitors of Viral Cell Entry

SARS-CoV-2 enters host cells by attaching to the cell membrane and subsequently fusing in the endosome. Entry requires the spike glycoprotein, which is cleaved by the host protease furin into S1 and S2 subunits during viral release from an infected cell. The subunits remain non-covalently associated and are present on mature virions as trimeric spikes. The S1 subunits bind the obligate SARS-CoV-2 receptor ACE2 with their RBDs, while the S2 subunits anchor the spike protein to the membrane and contain the fusion peptide. When S1 binds ACE2, an S2′ site within S2 is exposed and triggers its cleavage by the transmembrane serine protease 2 (TMPRSS2) at the cell surface, or by cathepsin L in the endosome after receptor-mediated endocytosis [153]. Every step in this intricate entry process is important and presents a potential DAA or HTA target. Clinical trials aimed at evaluating HTA are shown in Table 5.

Camostat mesylate, an inhibitor of TMPRSS2, has been reported as a potential entry inhibitor [154]. Although considered safe and well-tolerated, no clinical benefit was observed from RCTs results in terms of reduction of the disease progression or mortality (Table 5).

**Table 5 ijms-25-00354-t005:** Host-targeting antivirals (HTA) tested in clinical trials for COVID-19.

Drug Name	Initial Development Target	Mechanism	Trial Attributes	Clinical Trial Settings	Readout	Reference
Camostat mesylate	Chronic pancreatitis	Blocks ACE2 and RBD binding (TMPRSS2 inhibitor)	Prospective	A phase 1 study in healthy adult Japanese males in 2020.	Camostat mesylate was safe and tolerated at all dosages	[155]
Camostat mesylate	Chronic pancreatitis	Blocks ACE2 and RBD binding (TMPRSS2 inhibitor)	Prospective	An investigator-initiated, double-blind, randomised, placebo-controlled multi-centre trial in patients hospitalised with confirmed SARS-CoV-2 infection (n = 208) at 8 sites in Denmark in 2020.	No benefit on time to clinical improvement, progression to ICU admission or mortality.	[156]
Camostat mesylate	Chronic pancreatitis	Blocks ACE2 and RBD binding (TMPRSS2 inhibitor)	Prospective	A randomised, placebo-controlled phase 2 trial in asymptomatic (maximum 5 days) and asymptomatic patients with confirmed COVID-19 infection (n = 96) in Belgium in 2020.	Not effective as an antiviral drug against SARS-CoV-2.	[157]
Camostat mesylate	Chronic pancreatitis	Blocks ACE2 and RBD binding (TMPRSS2 inhibitor)	Prospective	A multi-centre, double-blind, randomised, parallel-group, placebo-controlled study in patients with mild to moderate COVID-19 with or without symptoms (n = 155) across 21 institutions in Japan in 2020 (CANDLE study).	No substantial reduction in the time to viral clearance, compared to the placebo.	[158]
Camostat monotherapy and favipiravir, camostat, and ciclesonide combination therapy	Chronic pancreatitis	Blocks ACE2 and RBD binding (TMPRSS2 inhibitor)	Prospective	A randomised, open-label, phase 3 study in hospitalised adults with moderate COVID-19 pneumonia (n = 121) in Japan from November 2020 to May 2021.	No significant differences in clinical findings between the groups.	[159]
Camostat mesylate	Chronic pancreatitis	Blocks ACE2 and RBD binding (TMPRSS2 inhibitor)	Prospective	A randomised, controlled, open-label, platform trial of adult patients hospitalised with COVID-19 pneumonia to receive either camostat mesylate or lopinavir/ritonavir (n = 121) from April 2020 to May 2021 in Vienna, Austria.	Shorter time to clinical improvement, reduced need for mechanical ventilation or death, and shorter length of stay than the use of lopinavir/ritonavir.	[160]
Camostat mesylate	Chronic pancreatitis	Blocks ACE2 and RBD binding (TMPRSS2 inhibitor)	Prospective	A double-blind, randomised, placebo-controlled, phase 2 study in adult patients with mild to moderate COVID-19 (n = 342) in South Korea in 2021.	No clinical benefit in patients with mild to moderate COVID-19.	[161]
Umifenovir (brand name arbidol)	Influenza virus	Inhibitor of virus entry	Prospective	Exploratory randomised controlled trial in patients with mild/moderate COVID-19 (n = 105) in Guangzhou, China in 2020.	Little benefit for improving the clinical outcome of patients compared to no antiviral control.	[63]
Umifenovir (brand name arbidol)	Influenza virus	Inhibitor of virus entry	Prospective	An open-label, randomised, controlled, single-centre trial where umifenovir was given following hydroxychloroquine compared to a control group where lopinavir/ritonavir was given following hydroxychloroquine. The study involved 100 patients in Iran in 2020.	Clinical and laboratory improvements were reported in the experiment group compared to the control group.	[162]
Umifenovir (brand name arbidol) combined with lopinavir	Influenza virus	Inhibitor of virus entry	Retrospective	A retrospective cohort study in adults aged ≥ 18 years with laboratory-confirmed COVID-19 without invasive ventilation (n = 33) in Guangdong, China in 2020.	Umifenovir in combination with lopinavir did not significantly improve clinical condition compared to lopinavir only.	[163]
Umifenovir (brand name arbidol)	Influenza virus	Inhibitor of virus entry	Prospective	Single-centre, randomised, open-label clinical trial in moderate-severe hospitalised patients with confirmed SARS-CoV-2 infection (n = 101) in Iran in 2020.	No effect in shortening the duration of viremia in severe patients, no improvement in the prognosis of non-ICU patients, and no reduction in mortality.	[164]
Hydroxychloroquine combined with or without azithromycin	Malaria parasite, rheumatoid arthritis, lupus, and porphyria	Inhibitor of virus entry	Prospective	A multi-centre, randomised, open-label, three-group, controlled trial in hospitalised patients with suspected or confirmed COVID-19 (n = 667) at 55 hospitals in Brazil in 2020.	No improvement in clinical status at 15 days as compared with standard care.	[165]
Hydroxychloroquine	Malaria parasite, rheumatoid arthritis, lupus, and porphyria	Inhibitor of virus entry	Prospective	A randomised, double-blind, placebo-controlled trial in asymptomatic adults (n = 821) who had household or occupational exposure to someone with confirmed COVID-19 in the United States and Canada in 2020.	No prevention of COVID-19 illness.	[166]
Hydroxychloroquine and azithromycin	Malaria parasite, rheumatoid arthritis, lupus, and porphyria	Inhibitor of virus entry	Prospective	A single-arm, open-label, non-randomised clinical trial in hospitalised patients (n = 36) in France in 2020.	Treatment is significantly associated with viral load reduction/disappearance in COVID-19 patients with a reinforced effect from the use of azithromycin.	[167]
Hydroxychloroquine	Malaria parasite, rheumatoid arthritis, lupus, and porphyria	Inhibitor of virus entry	Prospective	A randomised, controlled, open-label platform trial in patients hospitalised with COVID-19 (n = 1561) at 176 hospitals in the United Kingdom in 2020 (RECOVERY Collaborative Group study).	No clinical benefit in terms of lowering the incidence of death at 28 days compared to the usual care.	[168]
Hydroxychloroquine	Malaria parasite, rheumatoid arthritis, lupus, and porphyria	Inhibitor of virus entry	Prospective	A randomised trial with the intention to treat primary analysis in hospitalised patients with COVID-19 (n = 11,330) at 405 hospitals in 30 countries in 2020 (WHO SOLIDARITY trial).	Little or no effect is marked by no reduction in mortality, initiation of ventilation or hospitalization duration.	[66]
Hydroxychloroquine	Malaria parasite, rheumatoid arthritis, lupus, and porphyria	Inhibitor of virus entry	Prospective	A randomised, double-blind, placebo-controlled clinical trial in health care workers (n = 132) in Philadelphia and Pennsylvania, USA in 2020.	No clinical benefit was observed.	[169]
Hydroxychloroquine	Malaria parasite, rheumatoid arthritis, lupus, and porphyria	Inhibitor of virus entry	Prospective	An open-label, cluster-randomised trial involving asymptomatic contacts of patients with PCR-confirmed COVID-19 (n = 2314) in Spain in 2020.	No prevention of SARS-CoV-2 infection or symptomatic COVID-19 in healthy persons exposed to a PCR-positive case patient	[170]
Hydroxychloroquine	Malaria parasite, rheumatoid arthritis, lupus, and porphyria	Inhibitor of virus entry	Prospective	A randomised trial in critically ill patients (n = 2046) at 99 sites across 8 countries in 2020 (REMAP-CAP study).	Worsened outcomes compared to no antiviral therapy.	[68]
Hydroxychloroquine	Malaria parasite, rheumatoid arthritis, lupus, and porphyria	Inhibitor of virus entry	Prospective	A randomised, placebo-controlled trial in community-dwelling participants with confirmed SARS-CoV-2 infection (n = 148) in Alberta in 2020.	No evidence of the reduction of symptom duration or prevention of severe outcomes among outpatients with COVID-19.	[171]
Hydroxychloroquine combined with azithromycin	Malaria parasite, rheumatoid arthritis, lupus, and porphyria	Inhibitor of virus entry	Prospective	A randomised, placebo-controlled, double-blind, multi-centre trial in hospitalised patients with confirmed COVID-19 (n = 117) at 6 hospitals in Denmark in 2020 (Pro-PAC COVID study).	No improvement in survival or length of hospitalization in patients with COVID-19.	[172]
Hydroxychloroquine	Malaria parasite, rheumatoid arthritis, lupus, and porphyria	Inhibitor of virus entry	Prospective	A double-blind, randomised, placebo-controlled single-centre trial in health care workers caring for severe COVID-19 patients (n = 127) in Mexico in 2020.	No significant benefit compared to a placebo.	[173]
Disulfiram	Chronic alcoholism (aldehyde dehydrogenase inhibitor)	Inhibitor of virus entry	Retrospective	An observational study using a large database of clinical records of veterans (n = 944,127) in the United States between 2020 and 2021.	A possible contribution to reducing the incidence and severity of COVID-19.	[174]
Fluvoxamine	Antidepressant	FIASMA	Prospective	A double-blind, randomised, fully remote (contactless), placebo-controlled clinical trial in non-hospitalised adults with confirmed COVID-19 (n = 152) in Missouri and Illinois, USA in 2020.	Protective effects and lower likelihood of clinical deterioration compared to placebo on COVID-19 disease progression in outpatients.	[175]
Fluvoxamine	Antidepressant	FIASMA	Prospective	A multi-centre placebo-controlled, randomised, adaptive platform trial among high-risk symptomatic adults confirmed positive for SARS-CoV-2 with a known risk factor for progression to severe disease (n = 3323) at 11 clinical sites in Brazil in 2021 (TOGETHER study).	Reduction in risk of hospitalization or retention.	[176]
Fluvoxamine	Antidepressant	FIASMA	Prospective	An open-label, prospective cohort trial with matched controls in hospitalised patients with COVID-19 (n = 102) in Zagreb, Croatia in 2021.	Potential positive impact on patient survival.	[177]
Ivermectin	Helminth infection	Unknown exact mechanism	Prospective	A double-blind, randomised, placebo-controlled, adaptive platform trial in adult outpatients with early diagnosis of symptomatic SARS-CoV-2 infection (n = 3515) from 12 public health clinics in Brazil in 2020 (TOGETHER study).	No clinical benefit in lowering the incidence of COVID-19 disease progression.	[178]
Ivermectin	Helminth infection	Unknown exact mechanism	Prospective	An open-label, randomised, controlled adaptive platform trial in adults with early symptomatic COVID-19 (n = 224) in Thailand from September 2021 to April 2022 (PLATCOV study).	No measurable antiviral activity in early symptomatic COVID-19.	[179]
Nitazoxanide	Diarrhea-causing parasite	Inhibitor of viral replication (unknown)	Prospective	A randomised, double-blind, placebo-controlled pilot clinical trial in hospitalised individuals with mild respiratory insufficiency (n = 50) in Brazil in 2020.	Superior clinical and virological outcomes compared to placebo in time for hospital discharge, faster RT-PCR negativity, reduction of inflammatory markers, and reduction of lymphocyte T cells activation markers.	[180]
Nitazoxanide	Diarrhea-causing parasite	Inhibitor of viral replication (unknown)	Prospective	A multi-centre, randomised, double-blind, placebo-controlled trial in adult patients with early onset of disease (n = 392) in Brazil in 2020.	No difference in disease symptom resolution compared to the placebo group, although the reduction in viral load was observed.	[181]
Nitazoxanide	Diarrhea-causing parasite	Inhibitor of viral replication (unknown)	Prospective	A randomised, multi-centre, double-blind placebo-controlled clinical trial in outpatients with symptoms of mild to moderate COVID-19, enrolled within 72 h of symptom onset (n = 379) in the United States and Puerto Rico between August 2020 and February 2021.	No difference in time to sustained clinical recovery compared to a placebo	[182]
Nitazoxanide	Diarrhea-causing parasite	Inhibitor of viral replication (unknown)	Prospective	A phase 2, single centre, randomised. open-label in symptomatic adult outpatients (n = 192) in South Africa between September 2020 and August 2021.	No statistical difference in viral clearance for any drug regimen, compared to the standard of care.	[183]

ACE2, angiotensin-converting enzyme 2; FIASMA, Functional inhibitors of acid sphingomyelinase; RBD, receptor binding domain.

Umifenovir (arbidol) was developed for the treatment of influenza virus as a hemagglutinin inhibitor preventing virus-mediated fusion with the cell membrane, blocking viral entry into target cells [184,185]. Unsurprisingly, as SARS-CoV-2 viral entry does not depend on hemagglutinin, RCT results evaluating its activity against COVID-19 showed no favourable effect and limited efficacy of arbidol in treating COVID-19 [63,163] (Table 5, Figure 1).

Some inhibitors work by interacting with dynamin, a GTPase responsible for clathrin-dependent endocytosis, which is essential for coronaviruses to enter the cell [186]. Chlorpromazine (CPZ), an antipsychotic medication, has been repurposed as a COVID-19 therapeutic based on this mechanism and reported for its antiviral activity against MERS-CoV and SARS-CoV-1 [187]. These findings led to a clinical trial that observed a lower incidence of symptomatic COVID-19 among patients after treatment with CPZ [188].

Disulfiram is a hepatic aldehyde dehydrogenase inhibitor that is used to treat chronic alcoholism [174,189]. It reduced the incidence and severity of COVID-19 in a retrospective observational study (Table 5) [174], although no impact on viral load was shown [189]. Further large-scale clinical trials are needed to assess the findings.

Fluvoxamine is a psychotropic medication that belongs to a group of functional inhibitors of acid sphingomyelinase (FIASMA) which were evaluated against COVID-19 [190]. The rationale for using FIASMA is linked to the role of lipid rafts in viral entry. Sphingomyelinase activity is triggered by SARS-CoV-2 binding which in turn leads to the formation of ceramide-enriched membrane domains that help viral entry by clustering ACE2 [191]. Treatment with fluvoxamine inhibits the sphingomyelinase activity formation of these domains in vitro [191,192]. Fluvoxamine and other FIASMA drugs were associated with reduced mortality in COVID-19 patients were tested in a large cohort study [193]. Although three RCT studies showed favourable clinical benefits of fluvoxamine (Table 5), clinical evidence was deemed insufficient to issue a treatment recommendation in the National Institutes of Health (NIH) treatment guidelines [35].

### 3.2. Inhibitor of Viral Glycoprotein Folding

Enveloped viruses require the glycoprotein-folding machinery in the host endoplasmic reticulum (ER) to correctly fold their glycosylated proteins [40]. Pivotal players in this ER quality control (ERQC) pathway are the ER alpha glucosidases, which can be targeted for example by iminosugars [194,195,196]. Partially inhibiting the ERQC prevents the proper folding and incorporation of viral glycoproteins into budding viruses, as shown previously for other enveloped viruses such as HIV [197], human papillomavirus [198], dengue [196,199,200], influenza [196,201], hepatitis B virus [202], HCV [203], Zika virus [204], Marburg virus [205] and EBOV [206], and could lead to a potential broad-spectrum antiviral drug also against coronaviruses. The extensively glycosylated SARS-CoV-2 spike protein is essential for viral entry, and inhibition of its proper glycosylation leads to antiviral effects. The monocyclic UV-4 (N-(9-methoxynonyl)-1-deoxynojirimycin) or MON-DNJ prevented SARS-CoV-2-induced Vero cell death and reduced viral replication in vitro after 24 h of treatment [40,207]. The results are encouraging and need to be further tested in vivo and in clinical trials.

### 3.3. Host-Targeting Antivirals with Unknown Mechanism

Some repurposed drugs work by targeting the host, although the exact antiviral mechanisms are unknown. Ivermectin, an FDA-approved anti-parasitic drug was reported to have an in vitro antiviral activity to SARS-CoV-2 [208]. However, several adequately powered RCTs in Brazil [178], the US [209,210], and Malaysia [211] failed to report a clinical benefit from the use of ivermectin in COVID-19 outpatients and it is not approved or authorised by the FDA for the treatment of COVID-19.

Similar attention was given to the anti-malaria drug chloroquine and its derivatives [212]. Chloroquine was reported to have efficacy and acceptable safety against COVID-19-associated pneumonia in multi-centre clinical trials conducted in China [213] and its use attracted disproportionate attention during the coronavirus pandemic, spurred by preliminary studies and endorsement from political leaders [214]. The chloroquine derivative hydroxychloroquine was tested in RCTs with limited to no clinical benefit for COVID-19 (Table 5) [168]. Based on sufficiently powered randomised trials [169,215] however, NIH treatment guidelines recommend against the use of both chloroquine and hydroxychloroquine for the treatment of COVID-19 [35].

The anti-protozoal drug nitazoxanide also showed in vitro activity against SARS-CoV-2 [216], and has been tested in multiple RCTs (Table 5). A possible mechanism of action may be linked to inhibition of SARS-CoV-2 spike-induced syncytia and its binding to TMEM16 [217]. Nitazoxanide did not show efficacy in a number of RCTs when used at approved doses, and NIH treatment guidelines recommend against the use of nitazoxanide for the treatment of COVID-19 [35]. However, nitazoxanide did not show serious adverse events when evaluated in a Phase 1 study at higher doses of 1500 mg twice daily, at which it may provide antiviral efficacy according to the pharmacokinetic modelling [218].

## 4. Immunomodulatory Drugs

Immunomodulatory drugs are commonly used in autoimmune disease and include both monoclonal antibodies and small molecule inhibitors. They can either target cytokines directly, such as monoclonal antibodies against interleukins (ILs) or tumour necrosis factor (TNF)-alpha, inhibit proteins involved in inflammatory signalling pathways such as the janus kinase (JAK) inhibitors, or interfere with the hormonal regulation of inflammation such as corticosteroids [219,220]. Multiple monoclonal antibodies and immunomodulatory compounds were evaluated in COVID-19 infection (Table 6), driven by the aim to impact on COVID-19 associated systemic inflammation that can be associated with heightened cytokine release, as indicated by elevated blood levels of IL-6, C-reactive protein (CRP), D-dimer and ferritin [221,222,223,224]. Crucial for the assessment of the effects of immunomodulatory drugs were large-scale platform trials initiated early in the pandemic.

### 4.1. Corticosteroids

Corticosteroids bind to the glucocorticoid receptor and inhibit the synthesis of multiple inflammatory proteins through the suppression of genes that encode them, as well as promoting anti-inflammatory signals [224,225]. Coordinated from Oxford, the open-label RECOVERY trial recruited over 43,000 hospitalised patients with COVID-19 participants worldwide and randomly assigned patients to treatment groups that included dexamethasone, hydroxychloroquine, lopinavir-ritonavir, or azithromycin, and compared them to usual care, with the primary endpoint mortality at 28 days [226]. Dexamethasone treatment significantly decreased mortality in patients who were receiving either invasive mechanical ventilation or oxygen alone at randomization [14]. Another inhaled corticosteroid, budesonide, was assessed in the PRINCIPLE study in non-hospitalised patients with COVID-19, and improved recovery time with the potential to reduce hospital admissions or deaths [227].

### 4.2. Host-Targeting Monoclonal Antibodies

Multiple immunomodulatory monoclonal antibodies, directed against cytokines such as against tumour necrosis factor alpha (TNF-α), Interleukin (IL)-1, and IL-6, were assessed in COVID-19 patients [224]. Tocilizumab is a recombinant humanised monoclonal antibody that binds to interleukin-6 receptors, thereby blocking the activity of the pro-inflammatory cytokine. IL-6 is produced by a variety of cell types including lymphocytes, monocytes, and fibroblasts, and has been shown to be induced by SARS-CoV-2 infection in bronchial epithelial cells. The IL-6 inhibitor tocilizumab is in clinical use for several inflammatory diseases such as rheumatoid arthritis, giant cell arteritis, polyarticular juvenile idiopathic arthritis, and systemic juvenile idiopathic arthritis. As of 24 June 2021, the FDA has authorised the use of tocilizumab under EUA for the treatment of COVID-19 in hospitalised adults who are receiving systemic corticosteroids and require supplemental oxygen, mechanical ventilation, or extracorporeal membrane oxygenation (ECMO) [228]. The authorisation was based on the results of 4 clinical trials: RECOVERY, EMPACTA, COVACTA, and REMDACTA (Table 6), with the decision to grant EUA mainly based on the positive results of RECOVERY and EMPACTA that demonstrated an impact on mortality and a composite readout of mechanical ventilation and mortality [30,229].

IL-1 inhibitors include the IL-1 receptor antagonist Kineret (brand name anakinra), the IL-1 “Trap” rilonacept, and the neutralising monoclonal antibody to IL-1β canakinumab [224,230]. Anakinra was considered a safe and efficient treatment for severe forms of COVID-19 with a significant survival benefit in critically ill patients and features of macrophage activation-like syndrome [20,21], with most RCTs supporting the clinical benefit of this drug for COVID-19 (Table 6).

**Table 6 ijms-25-00354-t006:** Immunomodulatory drugs tested in clinical trials for COVID-19.

Drug Name	Initial Development Target	Mechanism	Trial Attributes	Clinical Trial Settings	Readout	Reference
Anakinra	Rheumatoid arthritis	IL-1 cytokine inhibitor	Prospective	A prospective, open-label, interventional study in adults hospitalised with severe COVID-19 pneumonia (n = 69) in Oman in 2020.	A significant reduction in inflammatory biomarkers and may confer clinical benefit.	[231]
Anakinra	Rheumatoid arthritis	IL-1 cytokine inhibitor	Prospective	A multi-centre, open-label, Bayesian randomised clinical trial in patients with mild to moderate COVID-19 pneumonia (n = 116) in France in 2020 (CORIMUNO-ANA-1 study).	No improved outcomes in patients with mild-to-moderate COVID-19 pneumonia compared to usual care.	[232]
Anakinra	Rheumatoid arthritis	IL-1 cytokine inhibitor	Prospective	An open-label trial in patients with COVID-19 (n = 130) in Greece in 2020.	A decreased risk of progression into severe respiratory failure.	[233]
Anakinra	Rheumatoid arthritis	IL-1 cytokine inhibitor	Prospective	A prospective, multi-centre, open-label, randomised, controlled trial, in hospitalised patients with COVID-19, hypoxia, and signs of a cytokine release syndrome (n = 342) in Belgium in 2020.	No reduction of the time to clinical improvement.	[234]
Anakinra	Rheumatoid arthritis	IL-1 cytokine inhibitor	Prospective	A double-blind, randomised controlled trial in patients with COVID-19 at risk of progressing to respiratory failure (n = 606) in Greece between Dec 2020 and Mar 2021 (SAVE-MORE study).	A decrease in the 28-day mortality and shorter duration of hospital stay.	[235]
Anakinra	Rheumatoid arthritis	IL-1 cytokine inhibitor	Prospective	An open-label prospective trial (n = 102) in Greece in 2020 (ESCAPE study).	Favorable responses among critically ill patients with COVID-19 and features of MALS.	[21]
Anakinra	Rheumatoid arthritis	IL-1 cytokine inhibitor	Prospective	An open-label, randomised, controlled trial in patients confirmed with COVID-19 (n = 30) in Iran in 2020.	Reduced need for mechanical ventilation and the hospital length of stay.	[236]
Anakinra	Rheumatoid arthritis	IL-1 cytokine inhibitor	Prospective	A controlled, open-label trial in adults with COVID-19 requiring oxygen (n = 71) at 20 sites in France in 2020 (ANACONDA study).	No efficacy compared to the standard of care.	[237]
Anakinra	Rheumatoid arthritis	IL-1 cytokine inhibitor	Prospective	An open-label, multi-centre, randomised clinical trial in patients with confirmed COVID-19 infection, evidence of respiratory distress, and signs of cytokine release syndrome (n = 80) in Qatar from October 2020 to April 2021.	No significant improvement compared to the standard of care.	[238]
Anakinra	Rheumatoid arthritis	IL-1 cytokine inhibitor	Prospective	A multi-centre, randomised, open-label, 2-group, phase 2/3 clinical trial in adult patients with severe COVID-19 pneumonia and hyperinflammation (n = 179) at 12 hospitals in Spain between May 2020 and March 2021 (ANA-COVID-GEAS study).	No reduction in the need for mechanical ventilation or reduction in mortality risk compared with standard of care alone.	[239]
Tocilizumab	Rheumatoid arthritis	IL-6 cytokine inhibitor	Prospective	A randomised, controlled, open-label multi-centre trial in COVID-19 (33 patients tocilizumab, 32 patients standard of care) at six hospitals in Anhui and Hubei, China in 2020.	Improvement of hypoxia in the tocilizumab group from day 12.	[32]
Tocilizumab	Rheumatoid arthritis	IL-6 cytokine inhibitor	Prospective	A randomised, placebo-controlled, trial in 389 patients hospitalised with COVID-19 pneumonia, randomised to tocilizumab (n = 249) and placebo (n = 128), enrollment from 6 countries in 2020 (EMPACTA).	Reduced likelihood of progression to the composite outcome of mechanical ventilation or death, but no improvement in survival.	[229]
Tocilizumab	Rheumatoid arthritis	IL-6 cytokine inhibitor	Prospective	A randomised, double-blind, placebo-controlled, multi-centre trial in 452 patients in Europe and North America, hospitalised with severe COVID-19 pneumonia, randomised to tocilizumab (n = 294) and placebo (n = 144) in 2020 (COVACTA trial).	Tocilizumab did not result in significantly better clinical status or lower mortality than placebo at 28 days.	[240]
Tocilizumab plus remdesivir	Rheumatoid arthritis	IL-6 cytokine inhibitor	Prospective	A randomised, double-blind, placebo-controlled, multi-centre trial included patients hospitalised with severe COVID-19 pneumonia requiring supplemental oxygen (REMDACTA trial), with 649 enrolled patients randomised to tocilizumab plus remdesivir (n = 434) and placebo plus remdesivir (n = 215) in Brazil, Russia, Spain, and the United States between June 2020 and March 2021.	Tocilizumab plus remdesivir did not shorten the time to hospital discharge or “ready for discharge” to day 28 compared with placebo plus remdesivir in patients with severe COVID-19 pneumonia.	[241]
Tocilizumab	Rheumatoid arthritis	IL-6 cytokine inhibitor	Prospective	A randomised, controlled, open-label, platform trial with 4116 patients in the United Kingdom with severe COVID pneumonia, randomised to tocilizumab and usual care (n = 2022), or usual care alone (n = 2094), enrolled between April 2020 and January 2021 (RECOVERY study).	Improved survival and other clinical outcomes	[30]
Tocilizumab and sarilumab	Rheumatoid arthritis	IL-6 cytokine inhibitor	Prospective	An international, multifactorial, adaptive platform trial in critically ill adult patients (n = 895) with COVID-19, randomised to tocilizumab (n = 353), sarilumab (n = 48), and standard care control (n = 402) in 2020 (REMAP-CAP study).	Improved clinical outcomes including 90-day survival of clinically ill patients after treatment with tocilizumab or sarilumab compared to the standard care.	[242]
Anakinra, tocilizumab, and siltuximab	Rheumatoid arthritis	IL-1 cytokine inhibitor (anakinra) IL-6 cytokine inhibitor (tocilizumab and siltuximab)	Prospective	A prospective, multi-centre, open-label, randomised, controlled trial in hospitalised patients (n = 342) with COVID-19, hypoxia, and signs of a cytokine release syndrome with a 2 × 2 factorial design to evaluate IL-1 blockade (n = 112) versus no IL-1 blockade (n = 230) and IL-6 blockade (n = 227; 114 for tocilizumab and 113 for siltuximab) versus no IL-6 blockade (n = 115) at 16 hospitals in Belgium in 2020 (COV-AID study).	Drugs targeting IL-1 or IL-6 did not shorten the time to clinical improvement.	[234]
Tocilizumab	Rheumatoid arthritis	IL-6 cytokine inhibitor	Prospective	A randomised, double-blind, placebo-controlled, multi-centre trial in patients hospitalised with COVID-19 (n = 452), assessed for efficacy and safety through day 60, in Europe and North America in 2020 (COVACTA study).	No benefit in reducing mortality up to day 60.	[243]
Adalimumab	Rheumatoid arthritis, Crohn’s disease	TNF-alpha cytokine inhibitor	Prospective	A randomised controlled trial in patients (n = 68) in Iran in 2020, where both the intervention and control groups received remdesivir, dexamethasone, and supportive care.	No significant difference between the two groups in terms of mortality rate and mechanical ventilation requirement. No effect on the length of hospital and ICU stay duration as well as radiologic changes.	[244]
Namilumab or infliximab	Rheumatoid arthritis	TNF-alpha cytokine inhibitor	Prospective	A randomised, multi-centre, multi-arm, multi-stage, parallel-group, open-label, adaptive, phase 2, proof-of-concept trial in hospitalised patients with COVID-19 pneumonia (n = 146) in the United Kingdom between June 2020 and February 2021 (CATALYST study).	Namilumab, but not infliximab, showed proof-of-concept evidence for a reduction in inflammation (measured as c-reactive protein/CRP concentration).	[245]
Infliximab	Rheumatoid arthritis	TNF-alpha cytokine inhibitor	Prospective	A randomised, multi-centre, double-masked, placebo-controlled clinical trial in hospitalised adult patients with COVID-19 pneumonia (n = 146) at 85 clinical research sites in the United States and Latin America between October 2020 and December 2021 (ACTIV-1 study).	No significant difference in time to recovery from COVID-19 pneumonia after treatment with abatacept, cenicriviroc, or infliximab, compared to placebo.	[246]
Dexamethasone	Arthritis and other inflammatory conditions	Corticosteroids	Prospective	A multi-centre, randomised, open-label, clinical trial in patients with COVID-19 and moderate to severe acute respiratory distress syndrome (n = 299) at 41 intensive care units in Brazil in 2020 (CoDEX study).	Significant increase in the number of ventilator-free days (days alive and free of mechanical ventilation) over 28 days, compared to standard care alone.	[247]
Dexamethasone	Arthritis and other inflammatory conditions	Corticosteroids	Prospective	A randomised, controlled, open-label trial (n = 2104) in the United Kingdom in 2020 (RECOVERY study).	Lower 28-day mortality among participants receiving either invasive mechanical ventilation or oxygen alone at randomization but not among those receiving no respiratory support.	[14]
Dexamethasone (high versus low dose)	Arthritis and other inflammatory conditions	Corticosteroids	Prospective	A multi-centre, randomised clinical trial in adults with confirmed COVID-19 requiring oxygen or mechanical ventilation (n = 1000) between August 2020 and May 2021 at 26 hospitals in Europe and India (COVID STEROID 2 study).	No statistically significant difference in more days alive without life support at 28 days.	[248]
Dexamethasone	Arthritis and other inflammatory conditions	Corticosteroids	Prospective	A multi-centre, placebo-controlled randomised clinical trial in adult patients admitted to the intensive care unit with COVID-19 (n = 546) at 19 sites in France between April 2020 to January 2021 (COVIDICUS study).	No significant improvement in 60-day survival.	[249]
Dexamethasone	Arthritis and other inflammatory conditions	Corticosteroids	Prospective	A randomised clinical trial between dexamethasone and methylprednisolone groups in hospitalised patients with COVID-19 (n = 143) in Iran in 2021.	Better effectiveness of dexamethasone compared with methylprednisolone.	[250]
Dexamethasone (high versus low dose)	Arthritis and other inflammatory conditions	Corticosteroids	Prospective	A randomised, open-label, controlled trial involving hospitalised patients with confirmed COVID-19 pneumonia needing oxygen therapy (n = 200) in Spain in 2021.	A high dose of dexamethasone reduced clinical worsening within 11 days after randomization, compared with a low dose.	[251]
Dexamethasone (high versus low dose)	Arthritis and other inflammatory conditions	Corticosteroids	Prospective	A multi-centre, randomised, open-label, clinical trial in patients with acute respiratory distress syndrome caused by COVID-19 (n = 100) in Argentina between June 2020 and March 2021.	No difference in the number of ventilator-free days between high versus low doses.	[252]
Budesonide	Ulcerative colitis	Corticosteroids	Prospective	A multi-centre, open-label, multi-arm, randomised, controlled, adaptive platform trial in outpatients (n = 4700) in the United Kingdom between November 2020 and March 2021 (PRINCIPLE study).	Improved time to recovery.	[227]
Budesonide	Ulcerative colitis	Corticosteroids	Prospective	A multi-centre, randomised, open-label trial in hospitalised patients with COVID-19 pneumonia (n = 120) in Spain and Argentina from April 2020 to March 2021 (TACTIC study).	Budesonide reduces the risk of disease progression.	[253]
Methylprednisolone versus dexamethasone	Arthritis and other inflammatory conditions	Corticosteroids	Prospective	A triple-blinded randomised controlled trial in hospitalised COVID-19 patients (n = 86) from August to November 2020 in Shiraz, Iran.	Methylprednisolone showed better efficacy than dexamethasone.	[254]
Methylprednisolone and dexamethasone	Arthritis and other inflammatory conditions	Corticosteroids	Prospective	A multi-centre, randomised, double-blind, placebo-controlled trial in hospitalised patients with COVID-19 pneumonia (n = 304) at 19 centres in Italy between December 2020 and March 2021.	No benefit in patients with COVID-19 pneumonia.	[255]
Baricitinib	Rheumatoid arthritis	Janus kinase inhibitor	Prospective	A multi-centre, phase 3, double-blind, randomised, placebo-controlled trial (n = 1525) at 101 centres across 12 countries in Asia, Europe, North America, and South America, between June 2020 and January 2021 (COV-BARRIER study).	No significant reduction in the frequency of overall disease progression.	[28]
Baricitinib plus remdesivir	Rheumatoid arthritis	Janus kinase inhibitor	Prospective	A double-blind, randomised, placebo-controlled trial in hospitalised adults with COVID-19 (n = 1033) at 67 trial sites in 8 countries: the United States, Singapore, South Korea, Mexico, Japan, Spain, the United Kingdom, and Denmark in 2020 (ACTT-2 study).	Combined with remdesivir, baricitinib was beneficial in reducing recovery time and accelerating improvement in clinical status, compared to remdesivir alone or placebo.	[26]
Baricitinib	Rheumatoid arthritis	Janus kinase inhibitor	Prospective	A randomised, controlled, open-label, platform trial in patients hospitalised with COVID-19 (n = 8156) in the United Kingdom in 2021 (RECOVERY study).	Reduced mortality in patients compared to standard of care alone.	[256]
Baricitinib versus dexamethasone	Rheumatoid arthritis	Janus kinase inhibitor	Prospective	A multi-centre randomised, double-blind, double placebo-controlled trial in hospitalised patients with COVID-19 requiring supplemental oxygen by low-flow, high-flow, or non-invasive ventilation (n = 1010) at 67 trial sites in the United States, South Korea, Mexico, Singapore, and Japan, between December 2020 and April 2021 (ACTT-4 study).	Similar mechanical ventilation-free survival by day 29 between groups, while dexamethasone was associated with significantly more adverse events, treatment-related adverse events, and severe or life-threatening adverse events.	[29]
Tofacitinib	Rheumatoid arthritis	Janus kinase inhibitor	Prospective	A randomised, placebo-controlled trial in patients hospitalised with COVID-19 (n = 289) at 15 sites in Brazil in 2020.	Lower risk of death or respiratory failure than placebo through day 28.	[257]
Tofacitinib	Rheumatoid arthritis	Janus kinase inhibitor	Prospective	An open-labeled randomised control study in hospitalised adult patients with mild to moderate COVID-19 pneumonia (n = 100) in India in 2020.	Reduction of the overwhelming inflammatory response compared to the standard care alone.	[258]

TNF-inhibitors have been used in severe cases of autoimmune inflammatory diseases such as rheumatoid arthritis, inflammatory bowel disease, or ankylosing spondylitis [259]. Several formulations of TNF inhibitors are currently available, including adalimumab, etanercept, and infliximab [224]. Elevated serum levels of TNF-α and soluble TNF-Receptor 1 have been detected in COVID-19 patients with severe infection, providing a rationale for the use of TNF-α inhibitors in SARS-CoV-2 infection [260]. However, concerns regarding the potential suppression of antiviral immune responses have been raised by an observational study showing a negative impact on SARS-CoV-2 antibody levels following natural infection in patients with inflammatory bowel disease treated with infliximab [261,262]. In the CATALYST open-label phase 2 trial, infliximab showed no impact on CRP levels as a measure for inflammation in SARS-CoV-2 infection [245]. Recently reported results of the ACTIV-1 trial, a placebo-controlled, masked, RCT among patients hospitalised for COVID-19, reported a significant benefit of infliximab on mortality [263], but no impact on the primary study endpoint length of pneumonia [246].

The latest addition to monoclonal antibodies under EUA is vilobelimab, a monoclonal antibody that specifically binds to the soluble human complement C5a, a product of complement activation. C5a activates the innate immune response, including the local release of histamines, contributing to inflammation and local tissue damage. In vivo studies demonstrated that an anti-C5a monoclonal antibody inhibited acute lung injury in a human C5a receptor knock-in mouse model [264]. The phase 3 PANAMO RCT study recorded an improvement in invasive mechanically ventilated patients’ survival that led to a decrease in mortality with the use of vilobelimab [19].

### 4.3. Janus Kinase (JAK) Inhibitors

The primary mechanism of action of Janus kinases (JAK) is the phosphorylating signal transducer and activator of transcription (STAT), a key player in signalling pathways involved signalling, growth, survival, inflammation, and immune activation. Inhibiting JAK prevents the phosphorylation of key signalling proteins involved in inflammation pathways, thereby blocking cytokine signalling [224,265]. Tofacitinib and baricitinib are orally available small-molecule JAK inhibitors approved for the treatment of rheumatoid arthritis [265] and have been evaluated in multiple RCTs in patients with SARS-CoV-2 infection, leading to a treatment recommendation in hospitalised adults that require respiratory support [35]. In brief, baricitinib had an impact on mortality [28,256], as well as reducing intensive care unit admissions, lowering the requirement for invasive mechanical ventilation, and improving patients’ oxygenation index [27,29], effects that were maintained in a meta-analysis [266]. Furthermore, baricitinib in combination with remdesivir showed superior results compared to remdesivir alone in reducing patients’ recovery time and improving their clinical status [26].

In summary, for immunomodulatory drugs, the NIH issued guidelines based on the existing evidence, recommending dexamethasone, tocilizumab, and baricitinib for hospitalised patients with COVID-19, whilst the evidence for anakinra, inhaled corticosteroids, and vilobelimab was deemed inconclusive, despite all compounds receiving EUA from the FDA.

## 5. Discussion

The devastating effect of the COVID-19 pandemic was a stark reminder of the requirement for antiviral compounds that are ready to use against viruses of pandemic potential and emerging viral threads.

A major accomplishment during the pandemic was the rapid implementation of several large platform trials that facilitated the evaluation of various repurposed compounds on COVID-19 hospitalisation and mortality, including DAA, HTA, and immunomodulatory drugs (Figure 3). Due to the time required to develop novel small molecule inhibitors, initial trials focused on immunomodulatory drugs and known antivirals developed against other viruses, followed by rapidly developed monoclonal antibodies to deliver treatments quickly [42,66,246,267]. Based on these RCTs, SARS-CoV-2 therapeutic management guidelines now include several repurposed drugs that were shown to be active against COVID-19. The success of initial repurposing efforts was highly variable for the different compound classes. Several immunomodulatory drugs such as baricitinib, tocilizumab, and dexamethasone improved SARS-CoV-2 disease severity and mortality and are now included in the treatment guidelines for hospitalised patients. Further, repurposing efforts of DAA—that were in the late stages of development but not yet approved at the start of the pandemic—shows the potential to rapidly translate in vivo antiviral activity findings into clinical trials [108]. This holds true for remdesivir and molnupiravir which were developed against the RdRp of RNA viruses EBOV and VEEV, respectively and are also highly effective against the SARS-CoV-2 RdRp. In contrast, repurposed DAA approved for HIV, HCV, or influenza, such as lopinavir, favipiravir, darunavir, and danoprevir, showed limited or no clinical activity against SARS-CoV-2 across RCTs. Not surprisingly, repurposed inhibitors of viral proteins that are not expressed by coronaviruses, such as the neuraminidase targeted by oseltamivir, did not show clinical activity [268]. Of note, comparably few repurposed HTA with a plausible mechanism of action were evaluated in large RCTs to date (Table 5).

Numerous in vitro high throughput screens have been conducted since early 2020 to identify potential candidates for repurposing, by screening approved and investigational drug collections [269,270]. However, many trials included compounds based on insufficiently validated cellular screening results, with extensive resources wasted in unnecessary investigations aiming to translate screening hits with marginal cellular activity or false positive data in vitro. This is particularly poignant for compounds causing phospholipidosis, a phenomenon caused by cationic amphiphilic drugs such as Chloroquine or Amiodarone leading to false positive results in vitro due to lipid processing inhibition [271]. Further, many compound libraries that were used in large in vitro screens, such as the ReFRAME library, are skewed towards human proteins, and therefore have, perhaps unsurprisingly, only yielded a few hits of compounds with previously known antiviral activity such as nelfinavir and MK-448 [272]. We, therefore, conclude that in vitro hits should be translated into clinical trials with caution, thoroughly reviewing in vitro and in vivo efficacy in relevant animal models, and, for DAA, establishing a robust pharmacokinetic/pharmacodynamic relationship and demonstrating consistent exposure over 90% effective concentration/EC_90_ prior to embarking on resource-demanding clinical trials.

With an abundance of in vitro screening data being generated, the sustainability and maintenance of data repositories is of particular importance, especially downstream for pandemic preparedness. The FAIR principles (Findable, Accessible, Interoperable, Reusable) provide a useful framework and should guide the development of data repositories that capture screening results against viruses of pandemic concern, which may aid pandemic response in the future. The maintenance of existing data repositories for cellular screening results, not only for SARS-CoV-2 but also for other viruses of pandemic concern, is of utmost importance to ensure an efficient pandemic response in the future.

In addition to repurposing efforts, the rapid discovery of novel, SARS-CoV-2 targeting DAA was unprecedented. This includes small molecule inhibitors such as M^Pro^ inhibitors nirmatrelvir, based on existing chemical starting points from previous coronavirus protease targets [25], or ensitrelvir, developed through de novo high-throughput screening hits that were progressed using SARS-CoV-2 specific fragment hits [50]. Similar to small molecule DAA efforts, highly efficient monoclonal antibodies designed to neutralise SARS-CoV-2 by binding to the spike protein on its surface were rapidly advanced early in the pandemic. However, many antibodies targeting spike lost their in vitro effectiveness against novel circulating variants such as Omicron and are no longer recommended for clinical use.

Drug resistance has always been the major challenge in drug development against viruses, including coronaviruses [273]. Newly evolving viral strains are linked to recurring waves, with partial immune escape and waning immunity within the population [274,275,276]. In addition, drug-induced viral mutations have been described for SARS-CoV-2 therapeutics, however, so far mostly in immunosuppressed individuals. Further, circulating viruses may harbour variants that are resistant to DAA, such as G15S and T21I that confer resistance to the M^Pro^ inhibitor Paxlovid [277,278,279]. The ongoing identification and characterisation of drug-resistant signatures within the SARS-CoV-2 genome will be crucial for clinical management and virus surveillance [280].

The accepted primary trial endpoints by the FDA include all-cause mortality, the need for hospitalisation and invasive mechanical ventilation, and a range of clinical signs such as sustained symptom alleviation [281]. In addition, a virological measure is acceptable as a primary endpoint in a Phase 2 clinical trial, but only as a secondary endpoint in a Phase 3 trial. However, over the last 3 years, circulating viral strains have changed significantly, and both vaccinations and natural infection have led to an increased immunological memory against SARS-CoV-2. Therefore, these endpoints are now less common, leading to discontinued RCTs such as for obeldesivir, due to lower-than-expected hospitalisations or mortality. This also leads to ethical considerations, on whether it is acceptable to enroll patients into a placebo-controlled trial where there is a very low risk of the primary endpoints of death or hospitalisation (e.g., <1%). A promising approach to overcome these issues may include pharmacodynamic modelling of viral clearance [282].

For pandemic preparedness, the concept of “one drug, multiple viruses” carries more promise than a “one drug, one virus” paradigm. Broad-spectrum antiviral agents that inhibit a wide range of human viruses should therefore be the target of de novo pandemic preparedness drug discovery efforts, as well as a focus on HTA that could be deployed immediately [283,284].

The 100-Day Mission set ambitious goals to prepare us for “Disease X”, aiming to generate safe, effective vaccines, therapeutics, and diagnostics within 100 days of the identification of a novel threat [285]. As well as developing novel assets, their licensing to ensure global and equitable access to future assets remains a key consideration. In the COVID-19 pandemic, the existing international laws and intellectual property practices failed to ensure equitable access to vaccines and therapeutics globally [286]. During the SARS-CoV-2 pandemic, companies appear to have financed their development efforts on the back of large procurement contracts with governments, rather than on the prospect of intellectual property, providing a useful case study for pandemic preparedness [287].

An urgent need remains [35,288] for drugs that can be easily stockpiled to ensure availability for pandemic preparedness.

## Figures and Tables

**Figure 1 ijms-25-00354-f001:**
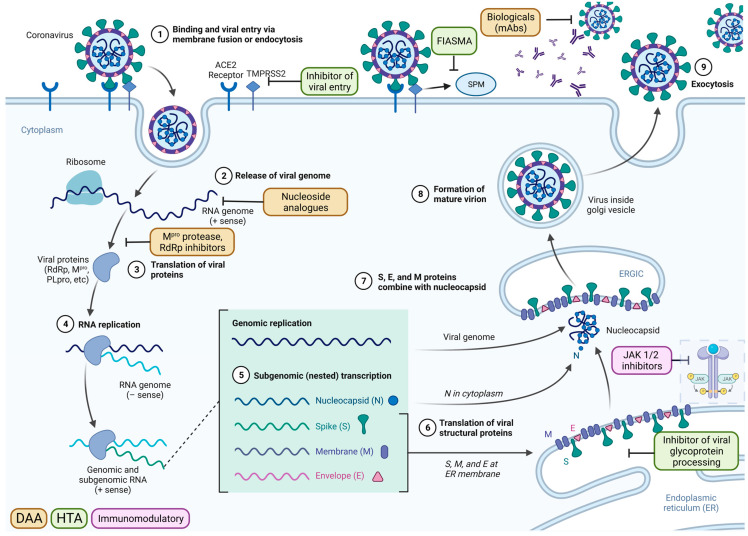
The coronavirus life cycle and the mechanistic actions of antiviral drugs within the viral replication process, using SARS-CoV-2 as an example. The virus-cell membrane fusion was induced by the binding of spike protein to the host cellular receptor angiotensin-converting enzyme 2 (ACE2), together with the cell surface transmembrane serine protease 2 (TMPRSS2). Following viral entry, the release of the viral genome is followed by the immediate translation of viral proteins and the formation of the viral replication and transcription complex. The 3-chymotrypsin-like protease (CL^pro^)/main protease (M^pro^) and papain-like protease (PL^pro^) cleave the virus polypeptide into 16 non-structural proteins. Structural glycoproteins are synthesised in the endoplasmic reticulum (ER) membrane for transit through the endoplasmic reticulum-to-Golgi intermediate compartment (ERGIC). Newly synthesised genomic RNA is encapsulated and buds into the ERGIC to form a virion. New virions leave the cell via lysosomes and are then able to infect new susceptible cells. SARS-CoV-2 infection activates the acid sphingomyelinase/ceramide system, resulting in the formation of ceramide-enriched membrane domains that serve viral entry and infection by clustering ACE2. The directly acting antivirals (DAA) mechanisms include the monoclonal antibodies that target the spike protein of the virus, M^pro^ inhibitor, nucleoside analogues, and RNA-dependent RNA polymerase (RdRp) inhibitor. The host-targeting antivirals (HTA) include the inhibitors of viral entry, functional inhibitors of acid sphingomyelinase activity (FIASMA), and inhibitors of viral glycoprotein processing. The immunomodulatory drugs modify the negative effects of an overreacting immune system, such as the interleukins and JAK ½ inhibitors. Adapted from “Coronavirus Replication Cycle”, by BioRender.com (2023). Retrieved from https://app.biorender.com/biorender-templates, accessed on 20 April 2023.

**Figure 2 ijms-25-00354-f002:**
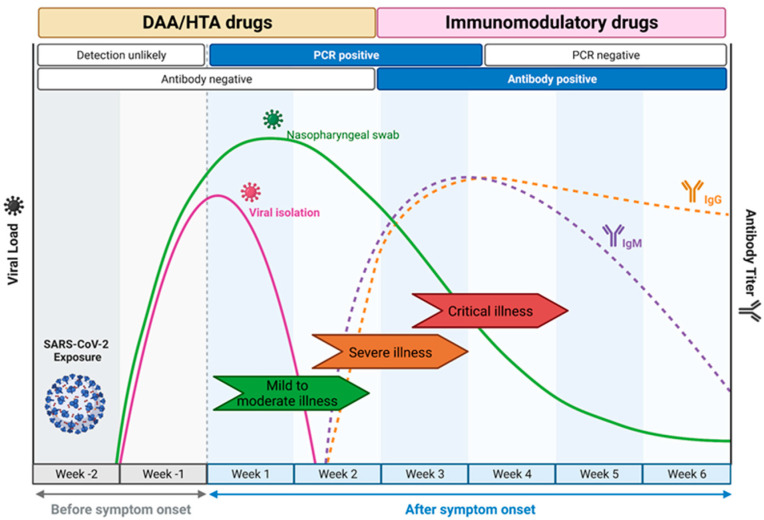
COVID-19 disease progression and the windows of opportunities where antiviral drugs should be deployed. The directly acting antivirals (DAA) and host-targeting antivirals (HTA) are most effective for an intervention in the earlier course of the mild to moderate disease manifestation when viral load is increasing and detectable by RT-PCR. The immunomodulatory drugs are more potent in the later phase of the disease when the host immune response starts to develop as a response to the infection and the clinical manifestation starts to develop from severe to critical illness due to the risks of a cytokine storm. Adapted from “Time Course of COVID-19 Infection and Test Positivity”, by BioRender.com (2023). Retrieved from https://app.biorender.com/biorender-templates, accessed on 20 April 2023.

**Figure 3 ijms-25-00354-f003:**
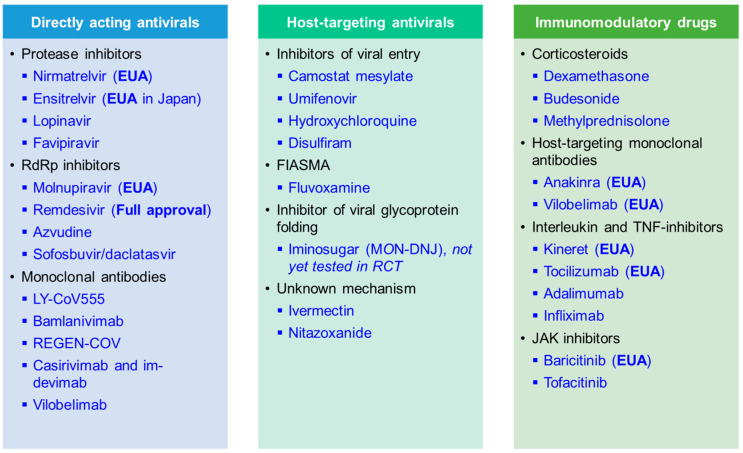
Summary of compounds discussed in this review article, including directly acting antiviral (DAA), host-targeting antiviral (HTA), and immunomodulatory drugs. Therapeutics that received full approval and emergency use authorisation (EUA) for COVID-19 are marked in the table. FIASMA, Functional inhibitors of acid sphingomyelinase activity.

**Table 1 ijms-25-00354-t001:** Coronaviruses with a history of pathogenicity in humans and their respective cellular entry receptors.

Virus Strain	Genus	Year of Discovery	Cellular Entry Receptors
HCoV-229E	alphacoronavirus	1966	Aminopeptidase N (APN)
HCoV-NL63	alphacoronavirus	2004	Angiotensin-converting enzyme 2 (ACE2)
HCoV-OC43	betacoronavirus	1967	9-*O*-acetylated sialic acid
HCoV-HKU1	betacoronavirus	2005	9-*O*-acetylated sialic acid
SARS-CoV-1	betacoronavirus	2003	ACE2
MERS-CoV	betacoronavirus	2012	Dipeptidyl peptidase 4 (DPP4)
SARS-CoV-2	betacoronavirus	2019	ACE2

**Table 2 ijms-25-00354-t002:** Biologicals and small molecule antiviral drugs granted full approval or emergency use authorisation (EUA) for the treatment of COVID-19 from the Food and Drug Administration (FDA) *.

Drug Name	Approval Status	Drug Description	Mechanism of Action	References
Remdesivir (brand name Veklury)	Full approval	RdRp inhibitor	DAA	[16,17]
Vilobelimab (brand name Gohibic)	EUA	Recombinant chimeric monoclonal IgG4 antibody	Immunomodulatory drug	[18,19]
Kineret (brand name Anakinra)	EUA	Interleukin-1 (IL-1) receptor antagonist	Immunomodulatory drugs	[20,21]
Molnupiravir (brand name Lagevrio)	EUA	Nucleoside analogue incorporating mutations in the RNA elongation process	DAA, RdRp inhibitor	[22,23]
Nirmatrelvir/ritonavir (brand name Paxlovid)	EUA	SARS-CoV-2 main protease (M^pro^) inhibitor	DAA, M^pro^ inhibitor	[24,25]
Baricitinib	EUA	JAK 1/2 inhibitor	Immunomodulatory drug	[26,27,28,29]
Tocilizumab (brand name Actemra)	EUA	IL-6 inhibitor	Immunomodulatory drug	[30,31,32,33]

DAA, directly acting antivirals; JAK, janus kinases; M^pro^, main protease; RdRp, RNA-dependent RNA polymerase. * Extracted and expanded from the FDA EUA list for Drugs and Non-Vaccine Biological Products at https://www.fda.gov/drugs/emergency-preparedness-drugs/emergency-use-authorizations-drugs-and-non-vaccine-biological-products [34], accessed 8 November 2023, and the NIH COVID-19 treatment guidelines [35] (Status 2 December 2023).

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
