# Peer review of "An Update on SARS-CoV-2 Clinical Trial Results—What We Can Learn for the Next Pandemic"

_ijms, 2023, doi:10.3390/ijms25010354_

Round 1
Reviewer 1 Report
Comments and Suggestions for Authors
This review is a good summary of the current knowledge of SARS-CoV-2 drugs. However, this reviewer has several comments to improve the manuscript.
Table1
Aren’t the receptors for HCoV-OC43 and HCoV-HKU “9-O-acetylated sialic acid”? (Fung & Liu, Annu Rev Microbiol. 2019. PMID: 31226023)
Abbreviations
There are many abbreviations, so it would be better to put a separate list of abbreviations. There are also many abbreviations that are not defined (e.g. SM inhibitor).
Clinical Trial Table
This reviewer thinks the authors should add period and country/region information of the clinical trial in the Clinical Trial Tables. This is because it gives us a clue as to what variants were prevalent during the clinical trial.
Minor Comments
Page 15, line 200: “EBV” is an abbreviation that appears only once and is generally Epstein-Barr Virus. Therefore, it should be removed.
Page 28, line 482: SARS-CoV-2 is spelled incorrectly.
Reviewer 2 Report
Comments and Suggestions for Authors
The article describes in great detail the recent and relevant information about COVID-19. For a better understanding of the article and the extensive information, it is suggested that you add 2 images, one that encompasses Host-targeting antivirals and another with immunomodulatory drugs.
